# Biased Dueling Bandits with Stochastic Delayed Feedback

**Bongsoo Yi**                                                                   *bongsoo@unc.edu*
*Department of Statistics and Operations Research*
*University of North Carolina at Chapel Hill*

**Yue Kang**                                                                      *yuekang@ucdavis.edu*
*Department of Statistics*
*University of California, Davis*

**Yao Li**                                                                        *yaoli@unc.edu*
*Department of Statistics and Operations Research*
*University of North Carolina at Chapel Hill*

**Reviewed on OpenReview:** *https://openreview.net/forum?id=HwAZDVxkLX*

## Abstract

The dueling bandit problem, an essential variation of the traditional multi-armed bandit problem, has become significantly prominent recently due to its broad applications in online advertising, recommendation systems, information retrieval, and more. However, in many real-world applications, the feedback for actions is often subject to unavoidable delays and is not immediately available to the agent. This partially observable issue poses a significant challenge to existing dueling bandit literature, as it significantly affects how quickly and accurately the agent can update their policy on the fly. In this paper, we introduce and examine the biased dueling bandit problem with stochastic delayed feedback, revealing that this new practical problem will delve into a more realistic and intriguing scenario involving a preference bias between the selections. We present two algorithms designed to handle situations involving delay. Our first algorithm, requiring complete delay distribution information, achieves the optimal regret bound for the dueling bandit problem when there is no delay. The second algorithm is tailored for situations where the distribution is unknown, but only the expected value of delay is available. We provide a comprehensive regret analysis for the two proposed algorithms and then evaluate their empirical performance on both synthetic and real datasets.

## 1 Introduction

In recent years, the dueling bandit problem has gained significant attention, finding wide applications in practical domains such as online advertising and recommendation systems (Yue & Joachims, 2009; Ailon et al., 2014; Zoghi et al., 2014a; 2015b; Komiyama et al., 2015; Agarwal et al., 2021). The *K-armed dueling bandit problem* (Yue et al., 2012) is a variation of the classical *K*-armed bandit problem (Auer et al., 2002), providing only relative comparison instead of absolute feedback. In the dueling bandit problem, the learner is presented with *K* arms. At each trial, the learner selects a pair of arms and receives a stochastic feedback indicating which of the two chosen options is preferred, based on an underlying stochastic pairwise preference model. This setup enables users to express their preference for one item over another rather than assigning numerical scores. Recognizing the challenges of consistently offering accurate absolute reviews, leveraging preference feedback with dueling bandits has emerged as a practical approach, gaining widespread popularity in online learning (Radlinski et al., 2008).

While existing studies in dueling bandit problem (Zoghi et al., 2014a; Komiyama et al., 2015) assume immediate feedback observation, a major practical challenge in real-world applications is the issue of delayed

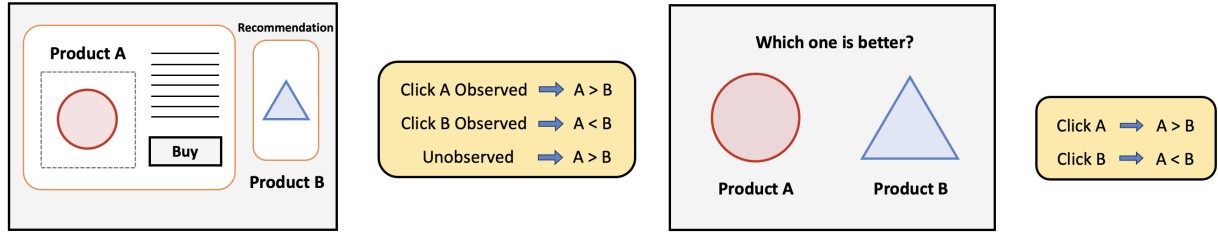

(a) Biased dueling bandit with delayed feedback        (b) Traditonal dueling bandit

Figure 1: Case (a) illustrates the e-commerce example of the biased dueling bandit with delayed feedback: a delay occurs before the user can evaluate and potentially select Product B, and any data collected during this delay suggests $A > B$. Case (b) showcases the traditional dueling bandit with immediate feedback.

feedback. In practice, feedback on actions is often not observable until after a random period. For example, in recommendations within e-commerce (Vernade et al., 2017), it takes time for users to decide whether to buy a product. Moreover, if a user has chosen not to purchase, the system remains unaware of this decision. It cannot distinguish whether the decision is not to buy or if the user has not made a decision yet, a dilemma often referred to as the *delayed conversion* problem. This complexity introduces significant challenges in managing delayed feedback. A similar situation occurs with web advertisements (Chapelle, 2014; Yoshikawa & Imai, 2018). Users take time to consider whether to click on an ad after viewing it. However, the system must continue to display ads to other users before receiving this feedback. In particular, Chapelle (2014) analyzed data from the online advertising company Criteo and confirmed that only 35% of feedback was observed within an hour, while 50% was observed after 24 hours, and notably, 13% emerged two weeks later. Another instance can be seen in clinical trials (Chow & Chang, 2006), where the impact of medical treatment on a patient's health frequently encounters delays. Due to these real-world challenges, the past few years have witnessed extensive research on various types of bandit problems with delayed feedback (Vernade et al., 2017; Grover et al., 2018). However, to the best of our knowledge, all the existing literature merely focuses on the bandit problems with absolute numerical rewards, and how to handle the delayed feedback under the practical dueling bandit problem receiving relative preferences between pairs still remains unexplored.

In this work, we introduce a new problem of dueling bandits in the presence of delayed feedback, characterized by a stochastic delay between the selection of action pairs and the receipt of corresponding preference feedback. Different from other types of bandits with delayed feedback, we further delve into a more practical and challenging problem called *biased dueling bandits under delayed feedback*. Specifically, we keep the same formulation as other bandit literature with binary rewards (Vernade et al., 2017; 2020): the player observes a zero value when the reward for an action is inaccessible due to delay. However, unlike other bandits with absolute numerical rewards where pulling one arm does not provide direct information about the others, the dueling bandit outputs binary responses revealing relative preference between pairs of arms. Consequently, the zero rewards under dueling bandits indicate a biased preference for one arm, particularly when rewards are delayed, making the delay effect much more malicious and challenging to handle. This treatment naturally gives an advantage to the second arm in a pair. For example, if arm $i$ is presented first and arm $j$ second, and the reward is delayed, the player observes a zero value while making a decision. This results in arm $j$ being favored, while arm $i$ suffers from the delay-induced bias. We will further elaborate on this issue in Section 2. In practice, this bias in our problem setting is very common and can be characterized as the prior knowledge or location effect indicating that one arm has an intrinsic advantage over the other. For instance, in the e-commerce recommendation shown in Figure 1 (a), a user is browsing product A, and the platform suggests product B on the sidebar. Before the user makes the comparison after some delay, we can only observe the user stay at product A and hence assume the user likes A over B. In other words, due to the common delay before the user can compare and perhaps select product B, any data collected during that delay may inaccurately suggest a preference for product A. This comparative bias exists in many real-world applications, and it highlights a unique and practical challenge in implementing dueling bandit algorithms with delayed feedback, compared to other bandit algorithms with delayed feedback where each

arm is evaluated based on independent selections and feedback, even if the feedback is delayed. Therefore, given the existing literature on bandits with delayed feedback, our work is highly novel and intriguing since we successfully address both the usual challenges of delayed feedback and the additional practical concerns of bias simultaneously.

Given the broad applicability of the dueling bandit problem and the significance of handling delayed feedback in real-world applications, our paper explores the dueling bandit with delayed feedback and presents two dueling bandit algorithms designed to handle stochastic delayed feedback. We note that the proposed algorithm can also be applied in general cases where there is no delay. Our main contributions can be summarized as follows:

- We describe and formulate a novel biased dueling bandit framework that incorporates stochastic delayed feedback.

- We present RUCB-Delay (Relative Upper Confidence Bound with delayed feedback), which deals with delayed feedback by employing the delay distribution information. We establish the regret bound for the algorithm and demonstrate that it matches the regret lower bound of the dueling bandit problem when all delays are zero.

- We introduce and analyze another algorithm, MRR-DB-Delay (Multi Round-Robin Dueling Bandit with delayed feedback), suitable for situations where the delay distribution is unknown. The algorithm requires only the expected value of the delay, making it versatile and applicable in various cases.

- We conduct an empirical evaluation of the performance of RUCB-Delay and MRR-DB-Delay using six synthetic and real-world datasets.

## 1.1 Related Work

Originating from the practical benefits of considering relative feedback over absolute feedback, the dueling bandit problem, initially introduced by Yue et al. (2009), has been widely investigated across various settings. Yue et al. (2009) began with strong assumptions, relying on strong stochastic transitivity and stochastic triangle inequality, which may frequently not align with real-world cases. Yue & Joachims (2011) suggested a setting on a relaxed form of strong stochastic transitivity, although this remained a fairly restrictive condition. In response, Urvoy et al. (2013) introduced the *Condorcet winner* setting, where we assume the existence of a unique best arm. Subsequent efforts have been made to generalize the concept of the Condorcet winner, including alternatives like *Borda winner* (Jamieson et al., 2015; Saha et al., 2021), *Copeland winner* (Zoghi et al., 2015a; Komiyama et al., 2016), and *Von-Neumann winner* (Dudík et al., 2015; Balsubramani et al., 2016). Nevertheless, our focus in this study remains on the Condorcet winner assumption, a premise commonly found in various other studies (Zoghi et al., 2014a;b; 2015b; Komiyama et al., 2015; Chen & Frazier, 2017; Haddenhorst et al., 2021; Agarwal et al., 2021; Saha & Gaillard, 2022).

Urvoy et al. (2013) and Zoghi et al. (2013) presented algorithms with a regret bound of $O(K^2 \log T)$ for the dueling bandit problem in the Condorcet winner setting. Subsequently, Zoghi et al. (2014a) and Komiyama et al. (2015) introduced RUCB and RMED algorithms, respectively, achieving a bound of $O(K^2 + K \log T)$ that matches a lower bound established in (Yue et al., 2012). RUCB is built on the estimation of pairwise probabilities and utilizes the Upper Confidence Bound (UCB) strategy to select the best arm, whereas RMED explores the likelihood of each arm being the Condorcet winner. Several other studies have examined the dueling bandit problem in specific settings, including investigations into robustness to corruptions (Agarwal et al., 2021), best-of-both-world scenarios (Saha & Gaillard, 2022), and adversarial settings (Gajane et al., 2015; Saha et al., 2021). To the best of our knowledge, our paper is the first to explore the dueling bandit problem with stochastic delayed feedback.

Delayed feedback, given its practical significance, has been a key area of research in multi-armed bandits and online learning (Joulani et al., 2013; Vernade et al., 2017; Grover et al., 2018; Gael et al., 2020; Lancewicki et al., 2021). A thorough analysis of online learning problems with delayed feedback including partial monitoring settings is given in Joulani et al. (2013). Additionally, Vernade et al. (2017) proposed algorithms

for delayed conversions in multi-armed bandits, assuming full knowledge of the delay distribution. Pike-Burke et al. (2018) addressed a more complex problem related to delayed, aggregated anonymous feedback. Here, the player only observes the sum of rewards received in each round after certain delays, without information about which past actions contributed to this aggregated reward. Delays have also been extensively explored in the adversarial setting (Cesa-Bianchi et al., 2016; Bistritz et al., 2019; Thune et al., 2019; Zimmert & Seldin, 2020). Furthermore, recent studies have investigated delays in various bandit settings and objectives, including linear bandits (Vernade et al., 2020), generalized linear bandits (Zhou et al., 2019; Howson et al., 2023), kernel bandits (Vakili et al., 2023), adversarial Markov decision processes (Lancewicki et al., 2022; Jin et al., 2022), and best-of-both-worlds algorithms (Masoudian et al., 2022; Saha & Gaillard, 2022).

## 1.2 Organization

The structure of the remaining sections in this paper is as follows: Section 2 provides a formal description of our problem setting. Following that, in Section 3, we introduce the RUCB-Delay algorithm, applicable when complete knowledge of the delay distribution is known. Section 4 presents another algorithm, MRR-DB-Delay, designed for scenarios where the delay distribution is unknown, with only the expected value known. The comprehensive regret analysis for the two algorithms, RUCB-Delay and MRR-DB-Delay, is also provided in Sections 3 and 4, respectively. In Section 5, we demonstrate the performance of our two algorithms using various simulated and real-world datasets. Finally, we conclude with a discussion of our results in Section 6.

## 2 Problem Setting

In this paper, we investigate a *dueling bandit* problem with $K$ arms, denoted as $\{1, 2, ..., K\}$. At each time step, we select a pair of arms $(i, j)$ and obtain the outcome of a pairwise comparison between the selected arms, which is subject to a delay. The outcome of the pairwise comparison between $(i, j)$ is stochastically determined by a probability $\mu_{ij}$, where $\mu_{ij}$ represents the likelihood of arm $i$ beating arm $j$ in a comparison between the two arms. The delay is decided by a discrete distribution $\mathcal{D}$, which is supported on $\mathbb{N}$ and is independent of the selection of arm pairs[1]. We assume the existence of a *Condorcet winner* (Urvoy et al., 2013), specifically arm 1 without loss of generality. This implies that there exists a unique arm, i.e. arm 1, with $\mu_{1i} = \mathbb{P}(1 \succ i) > \frac{1}{2}$ held for all $i \neq 1$.

In this work, we introduce a new problem of dueling bandits under the presence of delayed feedback. The exact procedure repeats the following steps:

1. At time step $t$, the player selects a pair of arms $(u_t, v_t)$.

2. The environment stochastically samples an outcome $X_t = \mathbb{I}(u_t \succ v_t) \in \{0, 1\}$ from the Bernoulli distribution $\mathcal{B}(\mu_{u_t v_t})$. Additionally, it determines a delay $D_t \in \mathbb{N}$ from the delay distribution $\mathcal{D}$.

3. At the beginning of time step $t + 1$, the player receives outcomes from earlier rounds. For $s < t + 1$, the censoring variable $\mathbb{I}(D_s \leq t + 1 - s)$ determines whether the outcome from time step $s$ is revealed by time step $t + 1$. Define $Y_{s,t+1} = X_s \mathbb{I}(D_s \leq t + 1 - s)$; the player then observes the collection $(Y_{s,t+1})_{1 \leq s \leq t}$.

Our problem setting aligns with the classic delayed feedback formulation with binary observations (Vernade et al., 2017; 2020): the player observes zero value for delayed responses until they become available. However, as we mention in Section 1, different from other types of bandits, the dueling bandit is intrinsically more difficult with delayed rewards due to its feedback mechanism that involves two arms at once: on the one hand, when $Y_{s,t} = 1$, it is straightforward to determine that $X_s = 1$. We say that the reward *converted* if $X_s = 1$ and the actual observation of this conversion event is indicated by $Y_{s,t} = 1$. On the other hand, if $Y_{s,t} = 0$, the status of $X_s$ ($X_s = 0$ or $X_s = 1$) remains uncertain due to the potential delay since we always

---

[1]We note that our analysis can be extended to scenarios where the delay distribution depends on the pair of arms. This is advantageous because a user may hesitate longer when $\mu_{ij}$ is close to $1/2$, resulting in longer delays. However, for simplicity and clarity, we assume independence in our work.

observe $Y_{s,t} = 0$ when the delayed feedback is not available yet regardless of the pairwise comparison result $X_s$. Specifically, the scenarios $X_s = 0$ or $D_s > t - s$ both would result in our observation $Y_{s,t} = 0$, making it impossible to distinguish between these two cases until the delay is resolved. Conclusively, this phenomenon leads to a preference bias on $v_t$ over $u_t$, and this issue is practical under the dueling bandit application as we mention in Section 1. Therefore, our problem is intrinsically more challenging than other types of bandits with delayed feedback due to the additional bias effects on pairwise comparisons.

Our goal is to minimize the cumulative regret up to time step $T$, given by

$$\mathcal{R}(T) = \sum_{t=1}^{T} \frac{\Delta_{u_t} + \Delta_{v_t}}{2}, \tag{1}$$

where $\Delta_i = \mu_{1i} - \frac{1}{2}$. This formulation aligns with the standard concept of regret used in other studies (Vernade et al., 2017; Pike-Burke et al., 2018; Vernade et al., 2020).

## 3 Algorithm: Known Delay Distribution

In this section, we introduce an algorithm designed for situations where the delay distribution $\mathcal{D}$ is known. Our approach begins with estimating the parameters $\mu_{ij}$ in the pairwise preference model. Following this, we propose RUCB-Delay, an algorithm adapted from the RUCB (Relative Upper Confidence Bound) algorithm (Zoghi et al., 2014a). RUCB-Delay utilizes the upper confidence bound (UCB) methodology computed from the estimated parameters. Finally, we establish a regret bound for our proposed algorithm, demonstrating its effectiveness in addressing the $K$-armed dueling bandit problem with delayed feedback.

### 3.1 Parameter Estimation

Let $\tau_t = \mathbb{P}(D_1 \leq t)$ represent the cumulative distribution function of the delay distribution. Assuming knowledge of the delay distribution, we possess information regarding the individual values of $\tau_t$. Additionally, we incorporate an $M$-threshold delay, restricting the observation of conversions to occur within $M$ time steps after the corresponding action. In other words, if the reward conversion has not been observed within the initial $M$ rounds, the algorithm presumes it will never occur. This *censored observation* setting aligns with the study conducted in Vernade et al. (2017; 2020). This assumption has a practical advantage, as there is no need to keep track of observations $\{Y_{s,t}\}$ for $t > s + M$.

We first introduce some key notations. We utilize the notation $\mathbb{I}_{i,j}^s = \mathbb{I}((u_s, v_s) = (i, j))$ for simplicity when there is no potential confusion in the context. Define $N_{ij}(t)$ as the exact count of times we select a pair of arms $(i, j)$ up to time $t$, and $\tilde{N}_{ij}(t)$ as the delay-discounted count, which takes into account the probability of the reward not yet being observed:

$$N_{ij}(t) = \sum_{s=1}^{t-1} \left( \mathbb{I}_{i,j}^s + \mathbb{I}_{j,i}^s \right), \tag{2}$$

$$\tilde{N}_{ij}(t) = \sum_{s=1}^{t-M} \tau_M \left( \mathbb{I}_{i,j}^s + \mathbb{I}_{j,i}^s \right) + \sum_{s=t-M+1}^{t-1} \tau_{t-s} \left( \mathbb{I}_{i,j}^s + \mathbb{I}_{j,i}^s \right). \tag{3}$$

Moreover, for practical benefit, as mentioned earlier, we introduce the censored observation $\tilde{Y}_{s,t}$, which differs from $Y_{s,t}$ only when the reward $X_s$ is one and the delay $D_s$ exceeds $M$:

$$\tilde{Y}_{s,t} = Y_{s,t} \, \mathbb{I}(D_s \leq M) = X_s \, \mathbb{I}(D_s \leq \min(M, t - s)).$$

Lastly, introduce $S_{ij}$ as follows:

$$S_{ij}(t) = \sum_{s=1}^{t-M} \tilde{Y}_{s,t} \, \mathbb{I}_{i,j}^s + \left( \tau_M - \tilde{Y}_{s,t} \right) \mathbb{I}_{j,i}^s + \sum_{s=t-M+1}^{t-1} \tilde{Y}_{s,t} \, \mathbb{I}_{i,j}^s + \left( \tau_{t-s} - \tilde{Y}_{s,t} \right) \mathbb{I}_{j,i}^s.$$

---

**Algorithm 1** RUCB-Delay

---

**Input:** Time horizon $T$, $\alpha$, $M$, $\{\tau_d\}_{d=1}^{M}$, $\mathcal{A} = \{1, 2, ..., K\}$
**Initialization:**

1: **for** $t = 1, 2, ..., T$ **do**
2:     Compute $u_{ij}(t)$ for all $i \neq j$ based on Equation 5
3:     $u_{ii}(t) \leftarrow \frac{1}{2}$ for all $i \in \mathcal{A}$
4:     Select a pair of arms $(u_t, v_t)$ according to RUCB
5:     Observe the collection $(Y_{s,t+1})_{1 \leq s \leq t}$.
6:     Update $N_{ij}(t+1)$, $\tilde{N}_{ij}(t+1)$, and $S_{ij}(t+1)$
7: **end for**

---

$S_{ij}(t)$ captures the bias-corrected count of arm $i$ beating arm $j$ up to time $t$. When comparing $(i, j)$ and if the current observation implies $j > i$, we add $\tau$ instead of 1. This adjustment accounts for the preference bias favoring the second arm over the first arm, which enables us to construct an unbiased estimator of $\mu_{ij}$. We define $\hat{\mu}_{ij}$ as follows:

$$\hat{\mu}_{ij}(t) = \frac{S_{ij}(t)}{\tilde{N}_{ij}(t)}, \tag{4}$$

which serves as a conditionally unbiased estimator of $\mu_{ij}$ when conditioned on the arm selections.

**Proposition 1.** *$\hat{\mu}_{ij}(t)$ is a conditionally unbiased estimator of $\mu_{ij}$, when conditioning on the selections of arms.*

*Proof.* If $(u_s, v_s) = (i, j)$,

$$\mathbb{E}(\tilde{Y}_{s,t}|u_s, v_s) = \mu_{ij}\tau_{\min(M, t-s)}.$$

Then,

$$\mathbb{E}(S_{ij}(t)|\{u_s, v_s\}_{1 \leq s \leq t-1})$$
$$= \sum_{s=1}^{t-M} \mu_{ij}\tau_M \, \mathbb{I}_{i,j}^s + (\tau_M - \mu_{ji}\tau_M) \, \mathbb{I}_{j,i}^s + \sum_{s=t-M+1}^{t-1} \mu_{ij}\tau_{t-s} \, \mathbb{I}_{i,j}^s + (\tau_{t-s} - \mu_{ji}\tau_{t-s}) \, \mathbb{I}_{j,i}^s = \mu_{ij}\tilde{N}_{ij}(t),$$

where the last equality holds from the definition of $\tilde{N}_{ij}(t)$ and the relation $\mu_{ji} = 1 - \mu_{ij}$. Therefore,

$$\mathbb{E}(\hat{\mu}_{ij}(t)|\{u_s, v_s\}_{1 \leq s \leq t-1}) = \mu_{ij}.$$

$\square$

### 3.2 Algorithm

Utilizing the estimator and variables defined in Section 3.1, we propose an algorithm named RUCB-Delay designed for the dueling bandit problem in the presence of delayed feedback. RUCB-Delay is built upon the framework of the RUCB algorithm (Zoghi et al., 2014a), retaining a similar structure but with novel estimators, variables, and an altered upper confidence bound. The modified upper confidence bound is given by:

$$U_{ij}(t) = \hat{\mu}_{ij}(t) + \sqrt{\frac{\alpha N_{ij}(t)\log t}{\tilde{N}_{ij}^2(t)}}, \tag{5}$$

where $\alpha \geq 1$. Refer to Algorithm 1 for an outline of the RUCB-Delay procedure. With the novel upper confidence bound, we follow RUCB's approach of selecting a pair of arms $(u_t, v_t)$. At each time step, we define a potential champion set $\mathcal{C}$ consisting of arms that optimistically win against all other arms, meaning $U_{ij} \geq \frac{1}{2}$ for all $j$. We then update the current best arm set $\mathcal{B}$, which will either have one element or be empty. The idea is twofold: first, the arm in $\mathcal{B}$ loses its top position as the best arm if it is optimistically beaten by another arm, and second, the arm $u_t$ will be selected from $\mathcal{B}$ with high probability, or from the potential

champion set $\mathcal{C}$. Specifically, if $\mathcal{C}$ contains exactly one element, we set $\mathcal{B}$ to $\mathcal{C}$ and select that unique element as $u_t$. If $\mathcal{C}$ has multiple elements, we choose $u_t$ from $\mathcal{B}$ with a probability of 0.5 and from the remaining potential champion arms in $\mathcal{C} \setminus \mathcal{B}$ with equal probability. After selecting the first arm $u_t$, the second arm $v_t$ is chosen as the one that maximizes $U_{u_t v_t}$.

### 3.3 Regret Analysis

Here, we present a regret analysis for the proposed RUCB-Delay algorithm. This is achieved by first establishing a general deviation inequality between $\mu_{ij}$ and $\hat{\mu}_{ij}$ in Lemma 2 and utilizing it to prove the high probability bound in Lemma 3.

**Lemma 2.** *For any $\alpha > 0$ and any pair of arms $(i, j)$, the following inequality holds for all $t$,*

$$\mathbb{P}\left(|\hat{\mu}_{ij}(t) - \mu_{ij}| > r_{ij}(t)\right) \leq \frac{2}{t^{2\alpha}} \quad \text{where } r_{ij}(t) = \sqrt{\frac{\alpha N_{ij}(t) \log t}{\tilde{N}_{ij}^2(t)}}.$$

The proof of Lemma 2 is provided in Appendix A.1. With the upper confidence bound $U_{ij}$ defined in Equation 5 for $i \neq j$ and $U_{ii} = \frac{1}{2}$ for all $i$, we define $L_{ij}(t) = 1 - U_{ji}(t)$. We now state the high probability concentration inequality:

**Lemma 3.** *Let $\alpha > \frac{1}{2}$ and $\delta > 0$. Then, with probability at least $1 - \delta$, for any $t > C(\delta)$ and any pair of arms $(i, j)$, the following holds:*

$$L_{ij}(t) \leq \mu_{ij} \leq U_{ij}(t),$$

*where $C(\delta) = \left(\frac{(4\alpha - 1)(M+1)K(K-1)}{(2\alpha - 1)\delta}\right)^{\frac{1}{2\alpha - 1}}$.*

The lemma establishes a high probability inequality that involves all large time steps and pairs of arms. By leveraging symmetry, our analysis can be confined to cases where $i < j$. Additionally, if arms $i$ and $j$ are compared at time $s_1$, then at time $s_2$, $N_{ij}$ and $\tilde{N}_{ij}$ remain constant for all time steps between $s_1 + M$ and $s_2$. This observation narrows down the range of time steps that need to be considered. Given these observations and Lemma 2, we present a comprehensive proof of Lemma 3 in Appendix A.1.

Given the high probability upper and lower bounds of $\mu_{ij}$ from Lemma 3, we derive the following high probability upper bound on the regret for the RUCB-Delay algorithm and state the expected regret bound.

**Theorem 1.** *Let $\alpha \geq 1$ and $\delta > 0$. For any $T \geq 1$, with probability at least $1 - \delta$, the cumulative regret $\mathcal{R}(T)$ of RUCB-Delay is upper bounded by*

$$O\left(\frac{MK^2}{\delta}\right) + \tilde{O}\left(\frac{1}{\tau_1^2}\sum_{i<j}\frac{\alpha}{\min(\Delta_i^2, \Delta_j^2)}\right) + O\left(\sum_{j=2}^{K}\frac{\alpha}{\tau_1^2 \Delta_j^2}\log T\right).$$

**Theorem 2.** *For $\alpha \geq 1$, the expected regret $\mathbb{E}[\mathcal{R}(T))]$ of RUCB-Delay is upper bounded by*

$$O\left(MK^2\right) + \tilde{O}\left(\frac{1}{\tau_1^2}\sum_{i<j}\frac{\alpha}{\min(\Delta_i^2, \Delta_j^2)}\right) + O\left(\sum_{j=2}^{K}\frac{\alpha}{\tau_1^2 \Delta_j^2}\log T\right).$$

The detailed proofs for Theorems 1 and 2 are available in Appendix A.2. The components of the expected regret upper bound in Theorem 2 share a similar structure with the regret bounds obtained in a fully stochastic setting (Zoghi et al., 2014a), but with an additional multiplicative constant $1/\tau_1^2$. We may easily verify that in the absence of delay, when $\tau_1 = 1$, our algorithm RUCB-Delay recovers the optimal regret bound in the standard stochastic dueling bandit problem (Zoghi et al., 2014a; Komiyama et al., 2015).

**Remark 1.** *We have developed an algorithm applicable to the delayed feedback setting in the dueling bandit problem by introducing a series of new variables and modifying the upper confidence bound when the delay distribution is known. We anticipate that many bandit algorithms using the Upper Confidence Bound (UCB)*

---
**Algorithm 2** Multi Round-Robin Dueling Bandit with Delayed Feedback (MRR-DB-Delay)

---
**Input:** Time horizon $T$, $\{n_m\}_{m \in \mathbb{N}}$
**Initialization:** $\gamma_1 = \frac{1}{2}, t = 1, m = 1, \mathcal{A}_1 = \{1, 2, ..., K\}$,
$T_{ij}(0) = \varnothing$ for all $i, j \in \mathcal{A}_1$

1: **while** $t \leq T$ **do**
2:      /* Round $m$ starts */
3:      **for** $i, j \in \mathcal{A}_m$ **do**
4:          Let $T_{ij}(m) := T_{ij}(m-1)$
5:          **while** $|T_{ij}(m)| \leq n_m$ and $t \leq T$ **do**
6:              Play arms $i$ and $j$
7:              $T_{ij}(m) \leftarrow T_{ij}(m) \cup \{t\}$
8:              $t \leftarrow t + 1$
9:          **end while**
10:      **end for**
11:      /* Update mean estimates and active arm set */
12:      **for** $i, j \in \mathcal{A}_m$ **do**
13:          $\bar{Y}_{ij}(m) := \frac{1}{|T_{ij}(m)|} \sum_{s \in T_{ij}(m)} Y_{s,t}$
14:      **end for**
15:      $\mathcal{A}_{m+1} := \mathcal{A}_m \setminus \{i \in \mathcal{A}_m : \exists j \in \mathcal{A}_m \text{ s.t. } \bar{Y}_{ij}(m) + \gamma_m < \frac{1}{2}\}$
16:      $\gamma_{m+1} \leftarrow \frac{\gamma_m}{2}$
17:      $m \leftarrow m + 1$
18: **end while**

---

*strategy can transition to scenarios with delayed feedback by employing novel variations similar to our upper bound.*

*For instance, the RR-DB algorithm (Saha & Gaillard, 2022) designed for stochastic dueling bandits can be transformed in a similar manner. RR-DB is a straightforward methodology that, in each round, compares all pairs of potentially optimal arms in a round-robin fashion, gradually removing significantly suboptimal arms. Although RR-DB achieves a regret bound of $O(K^2 \log T)$ and is not optimal overall, it performs optimally concerning the order of $\Delta_{\min}$. In contrast to other dueling bandit algorithms (Zoghi et al., 2014a; Bengs et al., 2021) suffering from a regret of order $\Delta_{\min}^{-2}$, RR-DB relies only on $\Delta_{\min}^{-1}$, which leads to an improved worst-case regret. We adjust the upper bound $u_{ij}$ of the RR-DB algorithm from*

$$u_{ij}(t) := \hat{p}_{ij}(t) + \sqrt{\frac{\log(Kt/\delta)}{N_{ij}(t)}}$$

*to*

$$u_{ij}(t) := \hat{\mu}_{ij}(t) + \sqrt{\frac{N_{ij}(t) \log(Kt/\delta)}{\tilde{N}_{ij}^2(t)}},$$

*where $\hat{\mu}_{ij}$ is defined as Equation 4. Then, the algorithm becomes suitable for the delayed feedback setting. We refer to this as RR-DB-Delay. Since RR-DB is not optimal, we do not delve into it in detail in this paper. However, we will evaluate and compare the performance of RR-DB-Delay in the experimental section (Section 5) later on.*

**Remark 2.** *We can formulate a non-asymptotic lower bound for the dueling bandit problem with stochastic delayed feedback. For any dueling bandit algorithm and $T$, there exists a $K$-armed dueling bandit such that $\mathcal{R}(T) \geq c\sqrt{TK/\tau_M}$, where $c > 0$ is a constant. The proof for this is deferred to Appendix D, while it remains an open problem whether this lower bound is tight.*

# 4 Algorithm: Known and Bounded Expected Delay

In practice, the exact delay distribution $\mathcal{D}$ is often unknown and can be challenging to estimate. This section presents a novel algorithm named Multi Round-Robin Dueling bandit with Delayed Feedback (MRR-DB-Delay) that can be implemented when only the expected value of the delay is known and bounded.

## 4.1 Algorithm

MRR-DB-Delay employs a round-based elimination strategy that iteratively refines the set of active arms. In each round, all possible pairs of active arms are played multiple times, and based on the accumulated observations, suboptimal arms are eliminated at the end of the round.

Algorithm 2 provides a detailed overview of our algorithm, MRR-DB-Delay. In this algorithm, $m$ denotes the current round, while $t$ represents the current time step, with $T$ being the total time steps until the algorithm concludes. Additionally, $n_m$ represents the number of times each active arm pair should be played by round $m$. In the collection $T_{ij}(m)$, we store all time steps when arm pair $(i, j)$ were played in the first $m$ rounds. The set $A_m$ identifies the active arms during round $m$, indicating the subset of arms currently under consideration.

In round $m$, each pair of arms $(i, j)$, where $i, j \in \mathcal{A}_m$, is played $n_m - n_{m-1}$ times. Then, we compute the mean estimates $\bar{Y}_{ij}(m)$ of arm pairs based on the observations at that point. Finally, arm $i$ is eliminated if there exists another arm $j$ in $\mathcal{A}_m$ such that the mean estimate $\bar{Y}_{ij}(m)$ is less than $\frac{1}{2}$ by $\gamma_m$. The gap tolerance $\gamma_m$ decreases exponentially over rounds, and in our further analysis (Lemma 4), we establish an inequality between the difference of $\bar{Y}_{ij}(m)$ and $\mu_{ij}$. These provide a guarantee that, with high probability, all arms, except the best arm 1, will be removed.

Lastly, in each round, we need to determine the value of $n_m$, indicating how many times we will play each pair of arms. The determination of $n_m$ is crucial as it plays a significant role in the algorithm's ability to efficiently eliminate suboptimal arms and has a substantial impact on regret performance. We choose $n_m$ so that the confidence bound established in Lemma 4 holds with high probability. This naturally results in the algorithm effectively removing arms with suboptimal performance. The selection of $n_m$ in Algorithm 4 is as follows:

$$n_m = \frac{C_1 \log(T\gamma_m^2)}{\gamma_m^2} + \frac{C_2 \sqrt{\mathbb{E}[D] \log(T\gamma_m^2)}}{\gamma_m} + \frac{C_3 \mathbb{E}[D]}{\gamma_m}, \tag{6}$$

where $C_1$ and $C_2$ are some constants, and $\mathbb{E}[D]$ represents the expected value of the delay. The complete formula for $n_m$ and further details on the derivation of $n_m$ are provided in Appendix B.1.

**Remark 3.** *In fact, with a different value of $n_m$, MRR-DB-Delay can also manage aggregated and anonymous delayed feedback (Pike-Burke et al., 2018). In this situation, the player only observes the reward summation in each round after some delay, without knowing which specific past actions led to this total reward. Detailed analysis on this aspect can be found in Appendix C.*

## 4.2 Regret Analysis

Here, we conduct a regret analysis for MRR-DB-Delay under the choice of $n_m$ in Equation 6. We establish an error bound between $\mu_{ij}$ and $\bar{Y}_{ij}(m)$, and subsequently derive the expected regret bound of the algorithm.

Let $t_m$ be the time step at the end of round $m$. The sum of discrepancies between $\mu_{ij}$ and the observation $Y_{s,t_m}$ during comparisons of arms $i$ and $j$ can be decomposed as follows:

$$\sum_{s \in T_{ij}(m)} (\mu_{ij} - Y_{s,t_m}) = \sum_{s \in T_{ij}(m)} (\mu_{ij} - X_s) + (X_s - Y_{s,t_m})$$

The first term represents the difference between the true parameter $\mu_{ij}$ and the outcome $X_s$, capturing the error in the absence of delayed feedback. Since the average of $X_s$ is an unbiased estimator of $\mu_{ij}$, the first term can be bounded using concentration inequalities. On the other hand, the second term is nonzero

only when the reward is converted but remains unobserved due to the delay, thereby illustrating the impact of delay on unobserved converted output. This decomposition allows us to separately analyze the effect of preference bias and delay. We establish an upper bound for each term and present the following high probability bound, along with a comprehensive proof given in Appendix B.1.

**Lemma 4.** *For any round $m \geq 1$ and any pair of active arms $(i, j) \in \mathcal{A}_m$, with probability at least $1 - \frac{2}{T\gamma_m^2}$,*

$$\mu_{ij} - \bar{Y}_{ij}(m) \leq \frac{\gamma_m}{2}.$$

We now provide the expected regret bound of MRR-DB-Delay in Theorem 3. Please refer to Appendix B.2 for a detailed proof.

**Theorem 3.** *The expected regret $\mathbb{E}[\mathcal{R}(T))]$ of Algorithm 2 is upper bounded by*

$$\sum_{i=2}^{K} O\left( \frac{K \log(T\Delta_i^2)}{\Delta_i} + K\sqrt{\log(T\Delta_i^2)\mathbb{E}[D]} + K\mathbb{E}[D] \right)$$

While its regret bound is larger than that of RUCB-Delay proposed in Section 3, MRR-DB-Delay possesses a significant advantage in that it can be utilized even when the complete delay distribution is unknown. Moreover, MRR-DB-Delay is advantageous in terms of the order of $\Delta_{\min}$, as it only depends on $\Delta_{\min}^{-1}$, whereas RUCB-Delay relies on $\Delta_{\min}^{-2}$.

## 5 Experiments

Following the introduction of our algorithms and theoretical analysis, this section involves a comparison of the performance of our proposed algorithms through numerical experiments conducted on various synthetic and real-world datasets.

**Algorithms.** We evaluate the performance of three algorithms introduced in our paper with the baseline: 1. RUCB-Delay (Section 3), 2. RR-DB-Delay (Remark 1 in Section 3), and 3. MRR-DB-Delay (Section 4). RUCB-Delay and RR-DB-Delay are algorithms that necessitate knowledge of the complete delay distribution, whereas MRR-DB-Delay only requires the expected value of the delay. We note that the regret bound, considering only terms related to $K$ and $T$, for RUCB-Delay is $O(K \log T)$, while RR-DB-Delay and MRR-DB-Delay have a bound of $O(K^2 \log T)$. To the best of our knowledge, there are no other known methods applicable to dueling bandit problems when there is a delay in feedback. However, we also included a baseline comparison using RUCB (Zoghi et al., 2014a) to effectively demonstrate the advantages of our proposed delay-specific algorithms. RUCB is an algorithm designed without accounting for delayed feedback; therefore, we update the output as soon as the delayed feedback is received.

**Experimental Setup.** We plot the regret performance of all three algorithms on six distinct datasets, which are described in the following paragraph. For all experiments, we set the time horizon to $T = 200,000$. However, for datasets where all three algorithms converge earlier than this fixed horizon, we plot the results only up to the earlier convergence time for better visualization. The regret performance for all plots was assessed by averaging the cumulative regret, as defined in Equation 1, across 100 runs. Both the average and standard deviation are reported in the plots. Similar to Vernade et al. (2017; 2020), we assume that the delay distribution follows a geometric distribution with $p = 0.01$, implying a mean $\mathbb{E}[D] = 100$. Also, based on our regret analysis in Theorem 2, we set $\alpha = 1.0$ for RUCB-Delay. For the baseline comparison, RUCB uses the number of times arm $i$ beats arm $j$ to establish their upper confidence bound, which includes potentially delayed feedback. We employ the most up-to-date values and update this count as soon as delayed feedback is received. Lastly, as discussed in Section 3.1, we introduce a windowing parameter denoted as $M$ for computational efficiency in RUCB-Delay and RR-DB-Delay. When dealing with an unbounded delay, we encounter practical storage issues since we need to keep track of previous actions to update $\tilde{N}_{ij}$ and $S_{ij}$. The windowing framework allows us to address this with an array of size $M$. We set the windowing parameter $M = 1000$. Given that $\tau_{1000} = 0.999$ with the geometric delay distribution, introducing the windowing parameter in this experiment is expected to have practically no impact on regret performance.

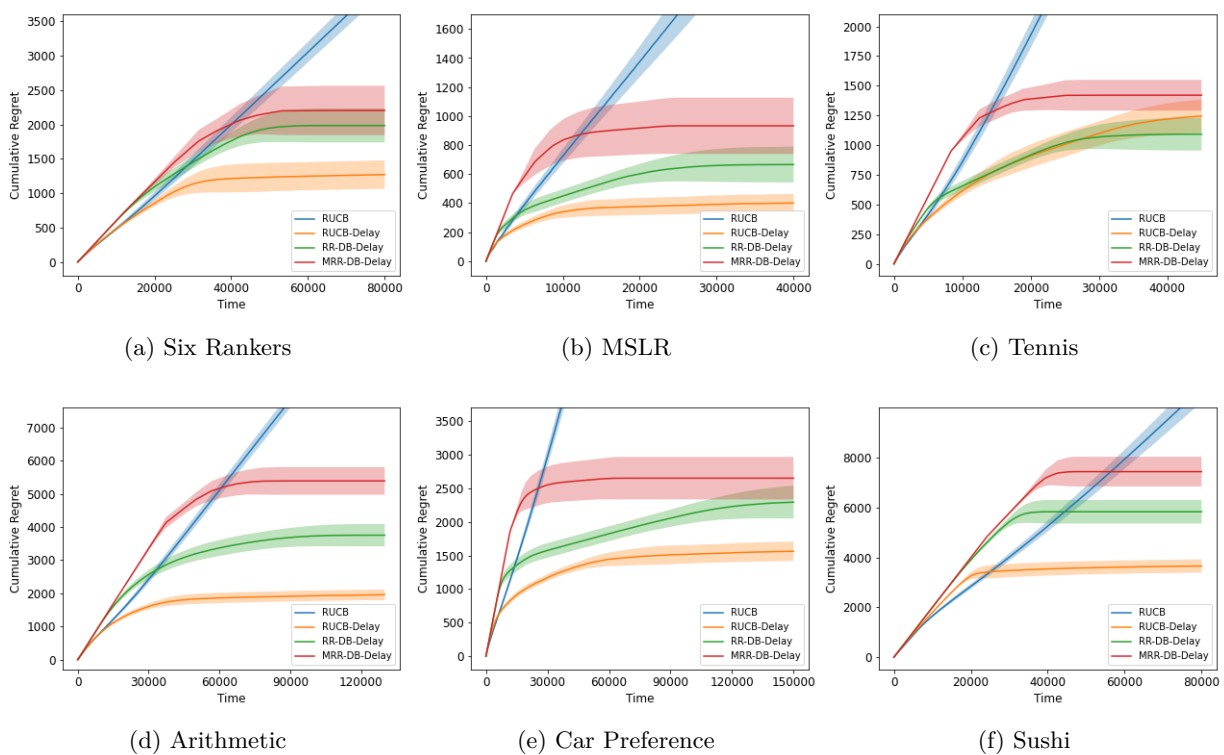

Figure 2: Average Cumulative Regret. The regret performances, averaged over 100 independent runs, along with their standard deviations, are reported for each dataset and algorithm.

**Datasets.** We perform experiments using the following six datasets. We provide a brief description of each dataset, including the number of arms:

- Six rankers ($K = 6$): a preference matrix generated from the six retrieval functions within the full-text search engine of ArXiv.org (Yue & Joachims, 2011).

- MSLR ($K = 5$): a $5 \times 5$ preference matrix introduced by Zoghi et al. (2015a) is extracted from a subset of rankers originating from the Microsoft Learning to Rank (MSLR) dataset (Qin & Liu, 2013). The MSLR dataset includes relevant information between queries and URLs, comprising more than 30,000 queries.

- Tennis ($K = 8$): a dataset, constructed by Ramamohan et al. (2016), is based on the results of tennis matches organized by the Association of Tennis Professionals (ATP) among 8 international tennis players. The element $(i, j)$ represents the proportion of times player $i$ has defeated player $j$.

- Arithmetic ($K = 10$): a synthetic preference matrix with ten arms where $\mu_{ij} = 0.5 + 0.025(j - i)$. A similar data was first used in (Komiyama et al., 2015).

- Car Preference ($K = 10$): a dataset of car preferences (Abbasnejad et al., 2013) collected from 60 users in the United States. The preference matrix was constructed based on the pairwise comparison data of users evaluating 10 different cars. Each user expressed their preferences for all 45 pairs of cars.

- Sushi ($K = 16$): a dataset derived from the sushi preference dataset (Kamishima, 2003), comprising the preferences of 5,000 Japanese users for 100 different types of sushi. Komiyama et al. (2015; 2016) selected 16 sushi types from the dataset and represented them in a preference matrix.

**Results.** The regret evaluations of the three algorithms for all datasets are presented in Figure 2. Across all datasets except for Tennis, RUCB-Delay outperforms RR-DB-Delay and MRR-DB-Delay, providing empirical support for their respective regret bound guarantees. Additionally, upon comparing the average performance of RR-DB-Delay and MRR-DB-Delay, we observe a slight superiority of RR-DB-Delay. However, it is essential to highlight that MRR-DB-Delay holds a notable advantage as it can be implemented with only the knowledge of the expected value of delay, unlike RUCB-Delay and RR-DB-Delay, which require complete delay distribution. Another critical aspect to highlight is the baseline comparison: The baseline method, RUCB, becomes futile and fails to converge across all datasets. This observation distinctly highlights the superiority of our approaches, specifically tailored for settings with delayed rewards. Notably, the innovative estimator introduced in RUCB-Delay significantly enhances the algorithm's effectiveness and robustness to delayed observations.

When examining the performance of RUCB-Delay, it can be observed that the regret increases more slowly in the early stages compared to RR-DB-Delay and MRR-DB-Delay. This fact is aligned with our expectation since both RR-DB-Delay and MRR-DB-Delay initially attempt all possible arm pairs before eliminating suboptimal arms, leading to a rapid increase in regret at the beginning. Additionally, for datasets with many arms having small $\Delta_i$, these arms tend to be eliminated later. As a result, comparably large differences in regret performance may arise between RUCB-Delay and both RR-DB-Delay and MRR-DB-Delay.

## 6 Discussion

We have studied the biased dueling bandit problem with stochastic delayed feedback and introduced two main algorithms, each accompanied by a comprehensive analysis. The first algorithm, RUCB-Delay, is designed for scenarios where the complete delay distribution is available. It leverages parameter estimation of the underlying model and incorporates a UCB strategy. This algorithm recovers the optimal regret bound for the dueling bandit problem in the absence of delay. The second algorithm, MRR-DB-Delay, is crafted for situations where information about the distribution is unavailable, and only the expected values are known. It employs a multi-round-robin fashion by playing all active arm pairs a predetermined number of times in each round. To the best of our knowledge, our methods are the first to delve into the delayed feedback setting within the dueling bandit framework. The efficiency of our proposed algorithms is then validated under numerical experiments.

There are several potential directions for future research. To begin with, it is intriguing to investigate algorithms that can be implemented without any knowledge of the delay, such as its expected value, while this problem remains challenging since existing algorithms for the simpler multi-armed bandit with delayed feedback still require additional side information. Furthermore, conducting theoretical studies to understand the impact of employing estimates of the delay distribution could be insightful. While our study assumed a Condorcet winner, it would be beneficial to extend the problem to consider Borda or Copeland winners. Last but not least, the biased dueling bandit presents a compelling area of study due to the pervasive nature of biases confounding user preferences in real-world applications. Despite its practical significance, this challenge remains largely underexplored in the existing literature.

### Acknowledgments

We thank the anonymous reviewers and the Action Editor for their constructive feedback, which helped improve this work. This work was supported in part by the National Science Foundation under grants DMS-2152289, DMS-2134107, and Cisco Faculty Award.

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

# A    Regret Analysis of RUCB-Delay

## A.1    Proof of Lemmas 2 and 3

**Lemma 2.** *For any $\alpha > 0$ and any pair of arms $(i, j)$, the following inequality holds for all $t$,*

$$\mathbb{P}\left(|\hat{\mu}_{ij}(t) - \mu_{ij}| > r_{ij}(t)\right) \leq \frac{2}{t^{2\alpha}} \quad \text{where } r_{ij}(t) = \sqrt{\frac{\alpha N_{ij}(t) \log t}{\tilde{N}_{ij}^2(t)}}.$$

*Proof.* Let

$$Z := \sum_{s=1}^{t-1} \tilde{Y}_{s,t} \, \mathbb{I}_{i,j}^s + (1 - \tilde{Y}_{s,t}) \, \mathbb{I}_{j,i}^s.$$

Then, we obtain the relation

$$S_{ij}(t) = Z + \sum_{s=1}^{t-1} (\tau_{\min(M, t-s)} - 1) \, \mathbb{I}_{j,i}^s.$$

Let's first consider the case conditioning on the selection of arms $\{u_s, v_s\}_{1 \leq s \leq t-1}$. Then, we have

$$S_{ij}(t) - Z = \sum_{s=1}^{t-1} (\tau_{\min(M, t-s)} - 1) \, \mathbb{I}_{j,i}^s = \mu_{ij} \tilde{N}_{ij}(t) - \mathbb{E}(Z).$$

Therefore, we can set an upper bound for the left-hand side conditioned on the selection of arms as:

$$\mathbb{P}\left(|\hat{\mu}_{ij}(t) - \mu_{ij}| > r_{ij}(t) \,|\, \{u_s, v_s\}_{1 \leq s \leq t-1}\right) = \mathbb{P}(|Z - \mathbb{E}(Z)| > r_{ij}(t) \tilde{N}_{ij}(t) \,|\, \{u_s, v_s\}_{1 \leq s \leq t-1}) \leq \frac{2}{t^{2\alpha}}.$$

The inequality holds by Hoeffding's Inequality since each censored observation $\tilde{Y}_{s,t}$ is independent given the selection of arms.

Finally, since the right-hand side term of the above inequality does not depend on the selection of arms, we may remove the conditioning by the law of total probability and obtain:

$$\mathbb{P}\left(|\hat{\mu}_{ij}(t) - \mu_{ij}| > r_{ij}(t)\right) \leq \frac{2}{t^{2\alpha}}.$$

$\square$

**Lemma 3.** *Let $\alpha > \frac{1}{2}$ and $\delta > 0$. Then, with probability at least $1 - \delta$, for any $t > C(\delta)$ and any pair of arms $(i, j)$, the following holds:*

$$L_{ij}(t) \leq \mu_{ij} \leq U_{ij}(t),$$

*where $C(\delta) = \left(\frac{(4\alpha-1)(M+1)K(K-1)}{(2\alpha-1)\delta}\right)^{\frac{1}{2\alpha-1}}$.*

*Proof.* Let $\mathcal{G}_{ij}(t)$ denote the *good* event where we have $\mu_{ij} \in [L_{ij}(t), U_{ij}(t)]$ at time $t$, and let $\mathcal{B}_{ij}(t)$ be the *bad* event where $\mu_{ij} \notin [L_{ij}(t), u_{ij}(t)]$ at time $t$.

For $i < j$, the relations $\mu_{ij} = 1 - \mu_{ji}$, $U_{ij}(t) = 1 - L_{ji}(t)$, and $L_{ij}(t) = 1 - U_{ji}(t)$ hold. Thus, the event $\mathcal{G}_{ij}(t)$ is true if and only if $\mathcal{G}_{ji}(t)$ is true. Additionally, $\mathcal{G}_{ii}(t)$ is always true because, as constructed, $\mu_{ii} = l_{ii}(t) = u_{ii}(t) = \frac{1}{2}$. Hence, it is sufficient to focus only on $\mathcal{G}_{ij}(t)$ for $i < j$.

We define $\sigma_n^{ij}$ as the time step when arms $i$ and $j$ were compared for the $n$-th time. We note that $N_{ij}(\sigma_n^{ij}) = n$ by definition. For any $n$, if $\sigma_n^{ij} + M < \sigma_{n+1}^{ij}$, then $\sqrt{\frac{\alpha N_{ij}(t) \log t}{\tilde{N}_{ij}^2(t)}}$ is an increasing function of $t$ for all $\sigma_n^{ij} + M \leq t < \sigma_{n+1}^{ij}$. Also, $\hat{\mu}_{ij}(t)$ is a constant for all $\sigma_n^{ij} + M \leq t < \sigma_{n+1}^{ij}$. Therefore, if $\mathcal{G}_{ij}(\sigma_n^{ij} + k)$ holds true for all $0 \leq k \leq M$, it implies that $\mathcal{G}_{ij}(t)$ holds for all $\sigma_n^{ij} \leq t < \sigma_{n+1}^{ij}$.

Using the above observations, for any $T > 0$, we have

$$\mathbb{P}\left(\forall t > T, i, j, \; \mathcal{G}_{ij}(t)\right)$$
$$= \mathbb{P}\left(\forall i < j, \; \mathcal{G}_{ij}(t) \text{ for } \forall T \leq t \leq \max\left(\sigma^{ij}_{N_{ij}(T)} + M, T\right)\right.$$
$$\left. \text{and } \mathcal{G}_{ij}(\sigma^{ij}_n + k) \text{ for } \forall n \geq N_{ij}(T) + 1, 0 \leq k \leq M\right).$$

Next, by considering the complement of the events on both sides of the above equation and applying the union bound, we arrive at the following:

$$\mathbb{P}\left(\exists t > T, i, j \text{ s.t. } \mathcal{B}_{ij}(t)\right)$$
$$\leq \sum_{i<j}\left(\mathbb{P}\left(\exists t \in \left[T, \max\left(\sigma^{ij}_{N_{ij}(T)} + M, T\right)\right] \text{ s.t. } \mathcal{B}_{ij}(t)\right)\right.$$
$$\left. + \mathbb{P}\left(\exists n \geq N_{ij}(T) + 1, 0 \leq k \leq M \text{ s.t. } \sigma^{ij}_n + k < \sigma^{ij}_{n+1} \text{ and } \mathcal{B}_{ij}(\sigma^{ij}_n + k)\right)\right)$$
$$\leq \sum_{i<j}\left(\underbrace{\mathbb{P}\left(\exists t \in \left[T, \max\left(\sigma^{ij}_{N_{ij}(T)} + M, T\right)\right] : |\mu_{ij} - \hat{\mu}_{ij}(t)| > \sqrt{\frac{\alpha N_{ij}(t) \log t}{\tilde{N}^2_{ij}(t)}}\right)}_{(a)}\right.$$
$$+ \underbrace{\mathbb{P}\left(\exists n \in [N_{ij}(T) + 1, T], k \in [0, M] : \sigma^{ij}_n + k < \sigma^{ij}_{n+1} \text{ and } \left|\mu_{ij} - \hat{\mu}_{ij}(\sigma^{ij}_n + k)\right| > \sqrt{\frac{\alpha N_{ij}(\sigma^{ij}_n + k) \log(\sigma^{ij}_n + k)}{\tilde{N}^2_{ij}(\sigma^{ij}_n + k))}}\right)}_{(b)}$$
$$\left. + \underbrace{\mathbb{P}\left(\exists n > T, k \in [0, M] : \sigma^{ij}_n + k < \sigma^{ij}_{n+1} \text{ and } \left|\mu_{ij} - \hat{\mu}_{ij}(\sigma^{ij}_n + k)\right| > \sqrt{\frac{\alpha N_{ij}(\sigma^{ij}_n + k) \log(\sigma^{ij}_n + k)}{\tilde{N}^2_{ij}(\sigma^{ij}_n + k))}}\right)}_{(c)}\right)$$

Let's break down the expression on the right-hand side into three components, denoted as $(a)$, $(b)$, and $(c)$, and then derive upper bounds for each of these components. The first part $(a)$ can be upper bounded as:

$$(a) \leq \mathbb{P}\left(\exists t \in [T, T + M] : N_{ij}(t) = N_{ij}(T) \text{ and } |\mu_{ij} - \hat{\mu}_{ij}(t)| > \sqrt{\frac{\alpha N_{ij}(t) \log t}{\tilde{N}^2_{ij}(t)}}\right)$$
$$\leq \mathbb{P}\left(\exists t \in [T, T + M] : N_{ij}(t) = N_{ij}(T) \text{ and } |\mu_{ij} - \hat{\mu}_{ij}(t)| > \sqrt{\frac{\alpha N_{ij}(t) \log T}{\tilde{N}^2_{ij}(t)}}\right)$$
$$\leq \sum_{k=0}^{M}\sum_{n=1}^{T}\mathbb{P}\left(N_{ij}(T + k) = N_{ij}(T) = n \text{ and } |\mu_{ij} - \hat{\mu}_{ij}(T + k)| > \sqrt{\frac{\alpha N_{ij}(T + k) \log T}{\tilde{N}^2_{ij}(T + k)}}\right)$$
$$\leq 2(M + 1)\sum_{n=1}^{T}\frac{1}{T^{2\alpha}}$$

where the last inequality holds from Lemma 2. Second, $(b)$ can be upper bounded as:

$$(b) \leq \mathbb{P}\left(\exists\, N_{ij}(T) + 1 \leq n \leq T,\, 0 \leq k \leq M : \sigma_n^{ij} + k < \sigma_{n+1}^{ij} \text{ and } \left|\mu_{ij} - \hat{\mu}_{ij}(\sigma_n^{ij} + k)\right| > \sqrt{\frac{\alpha N_{ij}(\sigma_n^{ij} + k)\log T}{\tilde{N}_{ij}^2(\sigma_n^{ij} + k))}}\right)$$

$$\leq \sum_{k=0}^{M}\sum_{n=1}^{T} \mathbb{P}\left(\sigma_n^{ij} + k < \sigma_{n+1}^{ij} \text{ and } \left|\mu_{ij} - \hat{\mu}_{ij}(\sigma_n^{ij} + k)\right| > \sqrt{\frac{\alpha N_{ij}(\sigma_n^{ij} + k)\log T}{\tilde{N}_{ij}^2(\sigma_n^{ij} + k))}}\right)$$

$$\leq 2(M+1)\sum_{n=1}^{T}\frac{1}{T^{2\alpha}}$$

The first inequality holds because $N_{ij}(T) + 1 \leq n$ and $\sigma_n^{ij} + k > T$, while the second inequality results from applying the union bound. The last inequality holds again from Lemma 2. Similarly, by using the fact that $n \leq \sigma_n^{ij}$ and Lemma 2, we can obtain an upper bound for $(c)$ as follows:

$$(c) \leq \mathbb{P}\left(\exists\, n > T,\, 0 \leq k \leq M : \sigma_n^{ij} + k < \sigma_{n+1}^{ij} \text{ and } \left|\mu_{ij} - \hat{\mu}_{ij}(\sigma_n^{ij} + k)\right| > \sqrt{\frac{\alpha N_{ij}(\sigma_n^{ij} + k)\log n}{\tilde{N}_{ij}^2(\sigma_n^{ij} + k))}}\right)$$

$$\leq \sum_{k=0}^{M}\sum_{n=T+1}^{\infty} \mathbb{P}\left(\sigma_n^{ij} + k < \sigma_{n+1}^{ij} \text{ and } \left|\mu_{ij} - \hat{\mu}_{ij}(\sigma_n^{ij} + k)\right| > \sqrt{\frac{\alpha N_{ij}(\sigma_n^{ij} + k)\log n}{\tilde{N}_{ij}^2(\sigma_n^{ij} + k))}}\right)$$

$$\leq 2(M+1)\sum_{n=T+1}^{\infty}\frac{1}{n^{2\alpha}}$$

Putting all these results together, we obtain

$$\mathbb{P}\left(\exists\, t > T, i, j \text{ s.t. } \mathcal{B}_{ij}(t)\right)$$

$$\leq 2(M+1)\sum_{i<j}\left(2\sum_{n=1}^{T}\frac{1}{T^{2\alpha}} + \sum_{n=T+1}^{\infty}\frac{1}{n^{2\alpha}}\right)$$

$$= 2(M+1)K(K-1)\frac{1}{T^{2\alpha-1}} + (M+1)K(K-1)\sum_{n=T+1}^{\infty}\frac{1}{n^{2\alpha}}$$

$$\leq 2(M+1)K(K-1)\frac{1}{T^{2\alpha-1}} + (M+1)K(K-1)\int_{T}^{\infty}\frac{dx}{x^{2\alpha}}$$

$$= \frac{(M+1)(4\alpha-1)K(K-1)}{(2\alpha-1)T^{2\alpha-1}}$$

Therefore, when $T = C(\delta)$, we obtain $\mathbb{P}\left(\exists\, t > C(\delta), i, j \text{ s.t. } \mathcal{B}_{ij}(t)\right) \leq \delta$ which concludes the proof. $\qquad\square$

## A.2  Proof of Theorems 1 and 2

**Theorem 1.** *Let $\alpha \geq 1$ and $\delta > 0$. For any $T \geq 1$, with probability at least $1 - \delta$, the cumulative regret $\mathcal{R}(T)$ of RUCB-Delay is upper bounded by*

$$O\left(\frac{MK^2}{\delta}\right) + \tilde{O}\left(\frac{1}{\tau_1^2}\sum_{i<j}\frac{\alpha}{\min(\Delta_i^2, \Delta_j^2)}\right) + O\left(\sum_{j=2}^{K}\frac{\alpha}{\tau_1^2\Delta_j^2}\log T\right).$$

*Proof.* We first state a high probability bound for the number of comparisons for each arm.

With probability at least $1 - \delta$, for any $t > C(\delta)$ and any pair of arms $(i, j) \neq (1, 1)$, the following holds:

$$N_{ij}(t) \leq \frac{4\alpha\log t}{\tau_1^2\min(\Delta_i^2, \Delta_j^2)}. \tag{7}$$

Moreover, let $N_{ij}^{\delta}(t)$ denote the number of comparisons between arms $i$ and $j$ performed between time $C(\delta)$ and $t$. Then,

$$N_{ij}^{\delta}(t) \leq \frac{4\alpha \log t}{\tau_1^2 \min(\Delta_i^2, \Delta_j^2)} \tag{8}$$

holds as well. The proof of this statement closely follows the analysis presented in Proposition 2 of Zoghi et al. (2014a), with a modification to Equation (3) of Zoghi et al. (2014a) as follows:

$$U_{ij}(s) - L_{ij}(s) = 2\sqrt{\frac{\alpha N_{ij}(s) \log s}{\tilde{N}_{ij}^2(s)}} \leq 2\sqrt{\frac{\alpha \log s}{\tau_1^2 N_{ij}(s)}} \leq 2\sqrt{\frac{\alpha \log t}{\tau_1^2 N_{ij}(t)}} \leq \min\{\Delta_i, \Delta_j\}.$$

Now, let $\hat{T}_{\delta}$ be the smallest time that such that

$$\hat{T}_{\delta} > C\left(\frac{\delta}{2}\right) + D \log \hat{T}_{\delta},$$

where $D := \frac{1}{\tau_1^2} \sum_{i<j} \frac{4\alpha}{\min(\Delta_i^2, \Delta_j^2)}$ and the existence of such $\hat{T}_{\delta}$ is guaranteed. Then, according to the above statement and Lemma 3, we can assert with a probability of $1 - \frac{\delta}{2}$ that

$$\forall t > C\left(\frac{\delta}{2}\right), i, j, \; \mu_{ij} \in [L_{ij}(t), U_{ij}(t)],$$

and

$$\forall (i,j) \neq (1,1), \; N_{ij}^{\delta/2}(\hat{T}_{\delta}) \leq \frac{4\alpha \log \hat{T}_{\delta}}{\tau_1^2 \min(\Delta_i^2, \Delta_j^2)}.$$

In this case, when $t > C\left(\frac{\delta}{2}\right)$ and $i > 1$, arm $i$ cannot be compared against itself. This is due to the fact that $u_{ii}(t) = \frac{1}{2} < \mu_{1i} \leq u_{1i}(t)$ which holds by Lemma 3. Additionally, we have

$$\left(\hat{T}_{\delta} - C\left(\frac{\delta}{2}\right)\right) - \sum_{i<j} N_{ij}^{\delta/2}(\hat{T}_{\delta}) \geq \left(\hat{T}_{\delta} - C\left(\frac{\delta}{2}\right)\right) - \frac{1}{\tau_1^2} \sum_{i<j} \frac{4\alpha \log \hat{T}_{\delta}}{\min(\Delta_i^2, \Delta_j^2)} > 0$$

by the definition of $\hat{T}_{\delta}$. Therefore, this implies that there exists a time $T_{\delta} \in \left(C\left(\frac{\delta}{2}\right), \hat{T}_{\delta}\right]$ when arm 1 was played against itself. At that time $T_{\delta}$, we had $u_{j1}(T_{\delta}) < \frac{1}{2}$ for all $j > 1$, indicating that $\mathcal{B} = \{1\}$.

With the above insights, we follow the proof strategy outlined in Theorem 4 of Zoghi et al. (2014a). Equation 7 in Zoghi et al. (2014a) requires modification, taking the form:

$$\tilde{N}_1(T) \leq \sum_{j=2}^{K} \tilde{N}_{1j}(T) \leq \sum_{j=2}^{K} \frac{4\alpha \log T}{\tau_1^2 \Delta_j^2} =: \hat{N}_1(T).$$

Furthermore, we can upper bound the cumulative regret similar to Equation 8 in Zoghi et al. (2014a) as follows:

$$\mathcal{R}(T) \leq T_{\delta} \Delta_{\max} + \sum_{j=2}^{K} \frac{4\alpha \Delta_{1j} \log T}{\tau_1^2 \Delta_j^2} + \sum_{l=1}^{\hat{N}_1(T)} n_l \Delta_{\max}. \tag{9}$$

While the notations for $C$ and $D$ have undergone modifications compared to Zoghi et al. (2014a), we may verify that all subsequent statements remain valid, and we are able to upper bound each term in Equation 9 as follows:

$$\mathcal{R}(T) \leq T_{\delta} \Delta_{\max} + \sum_{j=2}^{K} \frac{2\alpha \log T}{\tau_1^2 \Delta_j} + \sum_{l=1}^{\hat{N}_1(T)} n_l \Delta_{\max}$$

$$\leq \left(2C\left(\frac{\delta}{2}\right) + 2D \log 2D\right) \Delta_{\max} + \sum_{j=2}^{K} \frac{2\alpha \log T}{\tau_1^2 \Delta_j} + 4\Delta_{\max} \log \frac{2}{\delta} + \sum_{j=2}^{K} \frac{8\alpha}{\tau_1^2 \Delta_j^2} \Delta_{\max} \log T$$

which concludes the proof. $\qquad\square$

**Theorem 2.** *For $\alpha \geq 1$, the expected regret $\mathbb{E}[\mathcal{R}(T))]$ of RUCB-Delay is upper bounded by*

$$O\left(MK^2\right) + \tilde{O}\left(\frac{1}{\tau_1^2}\sum_{i<j}\frac{\alpha}{\min(\Delta_i^2, \Delta_j^2)}\right) + O\left(\sum_{j=2}^{K}\frac{\alpha}{\tau_1^2\Delta_j^2}\log T\right).$$

*Proof.* Let

$$H_T(1-\delta) := \left(2C\left(\frac{\delta}{2}\right) + 2D\log 2D\right)\Delta_{\max} + \sum_{j=2}^{K}\frac{2\alpha\log T}{\tau_1^2\Delta_j} + 4\Delta_{\max}\log\frac{2}{\delta} + \sum_{j=2}^{K}\frac{8\alpha}{\tau_1^2\Delta_j^2}\Delta_{\max}\log T.$$

For fixed $T$, we view $\mathcal{R}(T)$ as a random variable. By the relationship between the expected value of a random variable and its cumulative distribution function, we obtain:

$$\mathbb{E}[\mathcal{R}(T)] = \int_0^1 F_{\mathcal{R}(T)}^{-1}(q)dq < \int_0^1 H_T(q)dq.$$

Then, for integration of $H_T$, we only need to foucs on terms that depend on $\delta$:

$$\mathbb{E}[\mathcal{R}(T)] < \int_0^1 H_T(q)dq$$

$$= \int_0^1 \left(2C\left(\frac{1-q}{2}\right) + 2D\log 2D\right)\Delta_{\max} + \sum_{j=2}^{K}\frac{2\alpha\log T}{\tau_1^2\Delta_j} + 4\Delta_{\max}\log\frac{2}{1-q} + \sum_{j=2}^{K}\frac{8\alpha}{\tau_1^2\Delta_j^2}\Delta_{\max}\log T dq$$

$$= 2D\log 2D\Delta_{\max} + \sum_{j=2}^{K}\frac{2\alpha\log T}{\tau_1^2\Delta_j} + \sum_{j=2}^{K}\frac{8\alpha}{\tau_1^2\Delta_j^2}\Delta_{\max}\log T + \Delta_{\max}\int_0^1\left[2C\left(\frac{1-q}{2}\right) + 4\log\frac{2}{1-q}\right]dq.$$

Moreover, the above integral can be computed as:

$$\int_0^1\left[2C\left(\frac{1-q}{2}\right) + 4\log\frac{2}{1-q}\right]dq = \left(\frac{2(4\alpha-1)(M+1)K(K-1)}{2\alpha-1}\right)^{\frac{1}{2\alpha-1}}\frac{2\alpha-1}{\alpha-1} + 4(\log 2 + 1)$$

$$< \left(\frac{2(4\alpha-1)(M+1)K(K-1)}{2\alpha-1}\right)^{\frac{1}{2\alpha-1}}\frac{2\alpha-1}{\alpha-1} + 8.$$

Therefore, the expected regret can be upper bounded by

$$\mathbb{E}[\mathcal{R}(T))] \leq \left(8 + \left(\frac{2(4\alpha-1)(M+1)K(K-1)}{2\alpha-1}\right)^{\frac{1}{2\alpha-1}}\frac{2\alpha-1}{\alpha-1}\right)\Delta_{\max}$$

$$+ 2D\log 2D\Delta_{\max} + \sum_{j=2}^{K}\frac{2\alpha(\Delta_j + 4\Delta_{\max})}{\tau_1^2\Delta_j^2}\log T.$$

$\square$

**Remark 4.** *In studies focusing on delayed feedback (Vernade et al., 2017; 2020), certain analyses include $\tau_m$ in the regret bound instead of $\tau_1$. We note that in Theorem 1 and 2, $\tau_1$ can be replaced with $\frac{\tau_M}{M+1}$. This substitution is justified by the following inequality. If $N_{ij}(t-M) \geq 1$, then*

$$\tilde{N}_{ij}(t) \geq \frac{\tilde{N}_{ij}(t)}{M+1} + \frac{M}{M+1}\frac{\tilde{N}_{ij}(t)}{N_{ij}(t-M)}$$

$$\geq \frac{N_{ij}(t-M)}{M+1}\tau_M + \frac{M}{M+1}\tau_M$$

$$\geq \frac{\tau_M}{M+1}N_{ij}(t).$$

*The condition $N_{ij}(t-M) \geq 1$ is satisfied when all pairs of arms are initially selected before the beginning of the algorithm. Therefore, in further analysis, $\tau_1$ can be substituted with $\frac{\tau_M}{M+1}$. However, for simplicity, we maintain the use of $\tau_1$ in subsequent analysis.*

# B    Regret Analysis of MRR-DB-Delay

## B.1    Proof of Lemma 4

**Lemma 4.** *For any round $m \geq 1$ and any pair of active arms $(i,j) \in \mathcal{A}_m$, with probability at least $1 - \frac{2}{T\gamma_m^2}$,*

$$\mu_{ij} - \bar{Y}_{ij}(m) \leq \frac{\gamma_m}{2}.$$

*Proof.* Let $t_m$ denote the time step when round $m$ concludes, and define $P_s = X_s \mathbb{I}(D_s > t_m - s)\mathbb{I}(s \in T_{ij}(m))$. Additionally, denote $\mathcal{G}_t$ as the $\sigma$-algebra generated by actions, outcomes, delays, and observations up to time step $t$. We will start by proving two lemmas.

**Lemma 5.** *With probability greater than $1 - \frac{1}{T\gamma_m^2}$,*

$$\sum_{s \in T_{ij}(m)} (\mu_{ij} - X_s) \leq \sqrt{\frac{n_m \log(T\gamma_m^2)}{2}}.$$

*Proof.* Applying Hoeffding's inequality and Lemma 19 from Pike-Burke et al. (2018), we obtain the inequality:

$$\mathbb{P}\left(\sum_{s \in T_{ij}(m)} (\mu_{ij} - X_s) \geq \lambda\right) \leq \exp\left(\frac{-2\lambda^2}{n_m}\right)$$

Then, choosing $\lambda = \sqrt{\frac{n_m \log(T\gamma_m^2)}{2}}$ completes the proof. $\qquad\square$

**Lemma 6.** *Let $Q_t = \sum_{s=1}^{t} (P_s - \mathbb{E}(P_s|\mathcal{G}_{s-1}))$. With probability at least $1 - \frac{1}{T\gamma_m^2}$,*

$$Q_{t_m} \leq \frac{2}{3} \log(T\gamma_m^2) + \sqrt{2\mathbb{E}[D] \log(T\gamma_m^2)}$$

*Proof.* $\{Q_t\}_{t=0}^{\infty}$ is a martingale with respect to the filtration $\{\mathcal{G}_t\}_{t=0}^{\infty}$ with increments $Q_t - Q_{t-1} = P_t - \mathbb{E}(P_t|\mathcal{G}_{t-1})$. Also, we have

$$\sum_{s=1}^{t_m} \mathbb{E}\left(Q_s{}^2|\mathcal{G}_{s-1}\right) = \sum_{s=1}^{t_m} \text{Var}(P_s|\mathcal{G}_{s-1}) \leq \sum_{s=1}^{t_m} \mathbb{E}(P_s^2|\mathcal{G}_{s-1}) \leq \sum_{s=1}^{t_m} \mathbb{P}(D_s > t_m - s) \leq \mathbb{E}[D].$$

Then by Freedman's version of Bernstein's inequality (Theorem 1.6 of Freedman (1975)), we obtain the following inequality:

$$\mathbb{P}(Q_{t_m} \geq \lambda) \leq \exp\left(-\frac{\lambda^2/2}{\mathbb{E}[D] + \lambda/3}\right).$$

Finally, pick $\lambda = \frac{\log(T\gamma_m^2)}{3} + \sqrt{\frac{\log^2(T\gamma_m^2)}{9} + 2\mathbb{E}[D] \log(T\gamma_m^2)}$ and since

$$\frac{\log(T\gamma_m^2)}{3} + \sqrt{\frac{\log^2(T\gamma_m^2)}{9} + 2\mathbb{E}[D] \log(T\gamma_m^2)} \leq \frac{2}{3} \log(T\gamma_m^2) + \sqrt{2\mathbb{E}[D] \log(T\gamma_m^2)},$$

this concludes the proof. $\qquad\square$

Now, we employ the following decomposition:

$$\sum_{s \in T_{ij}(m)} (\mu_{ij} - Y_{s,t_m}) = \sum_{s \in T_{ij}(m)} (\mu_{ij} - X_s) + (X_s - Y_{s,t_m})$$

$$= \sum_{s \in T_{ij}(m)} (\mu_{ij} - X_s) + Q_{t_m} + \sum_{s=1}^{t_m} \mathbb{E}(P_s|\mathcal{G}_{s-1})$$

Both the first and second terms can be bounded with high probability using Lemma 5 and 6, respectively. The last term can be bounded as follows:

$$\sum_{s=1}^{t_m} \mathbb{E}(P_s|\mathcal{G}_{s-1}) \le \mu_{ij} \sum_{s=1}^{t_m} \mathbb{P}(D_s > t_m - s) \le \mathbb{E}[D]$$

Therefore, with a probability greater than $1 - \frac{2}{T\gamma_m^2}$, the following holds:

$$\mu_{ij} - \bar{Y}_{ij}(m) = \frac{1}{n_m} \sum_{s \in T_{ij}(m)} (\mu_{ij} - Y_{s,t_m}) \le \frac{2}{3n_m} \log(T\gamma_m^2) + \left( \frac{1}{\sqrt{2n_m}} + \frac{\sqrt{2\mathbb{E}[D]}}{n_m} \right) \sqrt{\log(T\gamma_m^2)} + \frac{\mathbb{E}[D]}{n_m} \quad (10)$$

Defining $n_m$ as

$$n_m = \left\lceil \frac{1}{\gamma_m^2} \left( \sqrt{\frac{\log(T\gamma_m^2)}{2}} + \sqrt{\frac{\log(T\gamma_m^2)}{2} + \frac{4}{3}\gamma_m \log(T\gamma_m^2) + 2\gamma_m \sqrt{2\mathbb{E}[D]\log(T\gamma_m^2)} + 2\gamma_m \mathbb{E}[D]} \right)^2 \right\rceil$$

ensures that the right-hand side of Equation 10 is less than or equal to $\frac{\gamma_m}{2}$, concluding the proof.

$\square$

## B.2 Proof of Theorem 3

**Theorem 3.** *The expected regret $\mathbb{E}[\mathcal{R}(T))]$ of Algorithm 2 is upper bounded by*

$$\sum_{i=2}^{K} O \left( \frac{K \log(T\Delta_i^2)}{\Delta_i} + K\sqrt{\log(T\Delta_i^2)\mathbb{E}[D]} + K\mathbb{E}[D] \right)$$

*Proof.* We follow the proof structure presented in the proof of Theorem 2 of Pike-Burke et al. (2018). For any arm $i$, let $M_i$ denote the round when arm $i$ is eliminated. Additionally, for any arm $i$, let's define

$$m_i := \min \left\{ m \ge 1 : \gamma_m < \frac{2}{3}\Delta_i \right\}.$$

Then by the definition of $m_i$, we have

$$\frac{\Delta_i}{3} \le \gamma_{m_i} < \frac{2}{3}\Delta_i.$$

Furthermore, we introduce $N_{ij}$ as the number of comparisons between arms $i$ and $j$ up to time $T$, and $N_i$ as the total number of times arm $i$ has been played by time $T$:

$$N_{ij} := N_{ij}(T) = \sum_{t=1}^{T} \mathbb{I}((u_t, v_t) = (i, j)) + \mathbb{I}((u_t, v_t) = (j, i))$$

$$N_i := \sum_{t=1}^{T} \mathbb{I}(u_t = i) + \mathbb{I}(v_t = i)$$

Also, for any arm $i$, let $\mathcal{R}_i(T) := N_i \frac{\Delta_i}{2}$. Then the cumulative regret can be expressed as

$$\mathcal{R}(T) = \sum_{i<j} N_{ij} \frac{\Delta_i + \Delta_j}{2} = \sum_{i=1}^{K} N_i \frac{\Delta_i}{2} = \sum_{i=1}^{K} \mathcal{R}_i(T).$$

Therefore, we decompose the expected regret as follows:

$$\mathbb{E}[\mathcal{R}(T)] = \mathbb{E}\left[\sum_{i=1}^{K} \mathcal{R}_i(T)\right]$$

$$= \underbrace{\mathbb{E}\left[\sum_{i=1}^{K} \mathcal{R}_i(T) \mathbb{I}(M_1 \geq m_i)\right]}_{(a)} + \underbrace{\mathbb{E}\left[\sum_{i=1}^{K} \mathcal{R}_i(T) \mathbb{I}(M_1 < m_i)\right]}_{(b)}$$

To bound each component, we prove the following two lemmas.

**Lemma 7.** *For any arm $i \neq 1$,*

$$\mathbb{P}(M_i > m_i \text{ and } M_1 \geq m_i) \leq \frac{2}{T\gamma_{m_i}^2}.$$

*Proof.* Define an event

$$A = \left\{ \bar{Y}_{i1}(m_i) \leq \mu_{i1} + \frac{\gamma_{m_i}}{2} \right\}$$

which occurs with high probability, specifically at least $1 - \frac{2}{T\gamma_{m_i}^2}$ by Lemma 4. If the event $A$ happens,

$$\bar{Y}_{i1}(m_i) \leq \mu_{i1} + \frac{\gamma_{m_i}}{2} = \frac{1}{2} - \Delta_i + \frac{\gamma_{m_i}}{2}$$
$$< \frac{1}{2} - \frac{3}{2}\gamma_{m_i} + \frac{\gamma_{m_i}}{2} = \frac{1}{2} - \gamma_{m_i}.$$

Therefore, if $M_1 \geq m_i$, we have $M_i \leq m_i$. This implies that if $M_i > m_i$ and $M_1 \geq m_i$, the event $A$ does not occur. Hence, we obtain

$$\mathbb{P}(M_i > m_i \text{ and } M_1 \geq m_i) \leq \mathbb{P}(A^c \cap \{i \in \mathcal{A}_{m_i}\}) \leq \frac{2}{T\gamma_{m_i}^2}.$$

$\square$

**Lemma 8.** *Let the event $F_i(m)$ be the event that the optimal arm $1$ is eliminated by arm $i$ in round $m$. Then, for any arm $i$,*

$$\mathbb{P}(F_i(m)) \leq \frac{2}{T\gamma_m^2}.$$

*Proof.* By the condition that an arm is eliminated,

$$\mathbb{P}(F_i(m)) = \mathbb{P}\left(1, i \in \mathcal{A}_m \text{ and } \bar{Y}_{1i}(m) + \gamma_m < \frac{1}{2}\right)$$

Define an event

$$A = \left\{ \bar{Y}_{1i}(m) > \mu_{1i} - \frac{\gamma_m}{2} \right\}.$$

If $A$ occurs,

$$\bar{Y}_{1i}(m) > \mu_{1i} - \frac{\gamma_m}{2} > \frac{1}{2} - \frac{\gamma_m}{2} > \frac{1}{2} - \gamma_m$$

which implies arm $1$ will not be removed by arm $i$ in round $m$. Therefore, by Lemma 4, we obtain

$$\mathbb{P}(F_i(m)) \leq \mathbb{P}(A^c \cap \{i \in \mathcal{A}_m\}) \leq \frac{2}{T\gamma_m^2}.$$

$\square$

**Bounding Term** $(a)$. By Lemma 7, we can bound the first term as follows:

$$\mathbb{E}\left[\sum_{i=1}^{K}\mathcal{R}_i(T)\mathbb{I}(M_1 \geq m_i)\right]$$

$$\leq \sum_{i=1}^{K}\mathbb{E}\left[\mathcal{R}_i(T)\mathbb{I}(M_i \leq m_i)\right] + \sum_{i=1}^{K}\mathbb{E}\left[\frac{T\Delta_i}{2}\mathbb{I}(M_i > m_i, M_1 \geq m_i)\right]$$

$$\leq \sum_{i=1}^{K}\frac{\Delta_i}{2}n_{m_i}K + \frac{T}{2}\sum_{i=1}^{K}\frac{2\Delta_i}{T\gamma_{m_i}^2}$$

$$\leq \sum_{i=1}^{K}\frac{\Delta_i}{2}n_{m_i}K + \sum_{i=1}^{K}\frac{9}{\Delta_i} = \sum_{i=1}^{K}\left(\frac{9}{\Delta_i} + \frac{\Delta_i n_{m_i}K}{2}\right).$$

**Bounding Term** $(b)$. Let $m_{\max} = \max_{i \neq 1} m_i$. By Lemma 7 and 8,

$$\mathbb{E}\left[\sum_{i=1}^{K}\mathcal{R}_i(T)\mathbb{I}(M_1 < m_i)\right]$$

$$= \sum_{m=1}^{m_{\max}}\mathbb{E}\left[\mathbb{I}(M_1 = m)\sum_{i:m_i>m}\frac{1}{2}N_i\Delta_i\right]$$

$$\leq \sum_{m=1}^{m_{\max}}\mathbb{E}\left[\mathbb{I}(M_1 = m)T\max_{i:m_i>m}\Delta_i\right]$$

$$\leq \sum_{m=1}^{m_{\max}}3\mathbb{P}(M_1 = m)T\gamma_m = \sum_{m=1}^{m_{\max}}\sum_{i=1}^{K}3\mathbb{P}(F_i(m))T\gamma_m$$

$$\leq \sum_{m=1}^{m_{\max}}\left(\sum_{i:m_i<m}3\mathbb{P}(M_1 \geq m_i, M_i > m_i)T\gamma_m + \sum_{i:m_i\geq m}3\mathbb{P}(F_i(m))T\gamma_m\right)$$

$$\leq \sum_{m=1}^{m_{\max}}\left(\sum_{i:m_i<m}\frac{6}{T\gamma_{m_i}^2}T\frac{\gamma_{m_i}}{2^{m-m_i}} + \sum_{i:m_i\geq m}\frac{6}{T\gamma_m^2}T\gamma_m\right)$$

$$\leq \sum_{i=1}^{K}\sum_{m=m_i}^{m_{\max}}\frac{6}{\gamma_{m_i}}2^{m_i-m} + \sum_{i=1}^{K}\sum_{m=1}^{m_i}\frac{6}{2^{-m}}$$

$$\leq \sum_{i=1}^{K}\frac{36}{\Delta_i} + \sum_{i=1}^{K}12\cdot 2^{m_i}$$

$$\leq \sum_{i=1}^{K}\frac{36}{\Delta_i} + \sum_{i=1}^{K}\frac{36}{\Delta_i} = \sum_{i=1}^{K}\frac{72}{\Delta_i}$$

where the last two inequalities hold from $2^{m_i} = \frac{1}{\gamma_{m_i}} \leq \frac{3}{\Delta_i}$.

Therefore, we have provided upper bound for terms $(a)$ and $(b)$, and the expected regret is overall upper bounded by:

$$\mathbb{E}[\mathcal{R}(T)] \leq \sum_{i=1}^{K}\left(\frac{81}{\Delta_i} + \frac{\Delta_i n_{m_i}K}{2}\right).$$

Lastly, our choice of $n_{m_i}$ can be bounded as follows:

$$
\begin{aligned}
n_{m_i} &= \left\lceil \frac{1}{\gamma_{m_i}^2} \left( \sqrt{\frac{\log(T\gamma_{m_i}^2)}{2}} + \sqrt{\frac{\log(T\gamma_{m_i}^2)}{2} + \frac{4}{3}\gamma_{m_i}\log(T\gamma_{m_i}^2) + 2\gamma_{m_i}\sqrt{2\mathbb{E}[D]\log(T\gamma_{m_i}^2)} + 2\gamma_{m_i}\mathbb{E}[D]} \right)^2 \right\rceil \\
&\leq \left\lceil \frac{1}{\gamma_{m_i}^2} \left( 2\log(T\gamma_{m_i}^2) + \frac{8}{3}\gamma_{m_i}\log(T\gamma_{m_i}^2) + 4\gamma_{m_i}\sqrt{2\mathbb{E}[D]\log(T\gamma_{m_i}^2)} + 4\gamma_{m_i}\mathbb{E}[D] \right) \right\rceil \\
&\leq 1 + \frac{2\log(4T\Delta_i^2/9)}{\gamma_{m_i}^2} + \frac{8\log(4T\Delta_i^2/9)}{3\gamma_{m_i}} + \frac{4\sqrt{2\mathbb{E}[D]\log(4T\Delta_i^2/9)}}{\gamma_{m_i}} + \frac{4\mathbb{E}[D]}{\gamma_{m_i}} \\
&\leq 1 + \frac{18\log(4T\Delta_i^2/9)}{\Delta_i^2} + \frac{8\log(4T\Delta_i^2/9)}{\Delta_i} + \frac{12\sqrt{2\mathbb{E}[D]\log(4T\Delta_i^2/9)}}{\Delta_i} + \frac{12\mathbb{E}[D]}{\Delta_i}
\end{aligned}
$$

Thus, the total expected regret is bounded by

$$
\mathbb{E}[\mathcal{R}(T)] \leq \sum_{i=2}^{K} \left( \frac{9K\log(4T\Delta_i^2/9)}{\Delta_i} + 4K\log(4T\Delta_i^2/9) + 6K\sqrt{2\mathbb{E}[D]\log(4T\Delta_i^2/9)} + \frac{81}{\Delta_i} + 6K\,\mathbb{E}[D] + \frac{1}{2}K\Delta_i \right)
$$

$\square$

## C MRR-DB-Delay on delayed, aggregated, and anonymous feedback

In this section, we illustrate the regret guarantee of MRR-DB-Delay when handling delayed, aggregated, and anonymous feedback in the dueling bandit problem. Here we also assume a known and bounded delay as in Section 4. For a comprehensive description of the delayed, aggregated, and anonymous feedback setting, please refer to Pike-Burke et al. (2018). We conduct a regret analysis with the following choice of $n_m$:

$$
n_m = \left\lceil \frac{1}{\gamma_m^2} \left( \sqrt{2\log(T\gamma_m^2)} + \sqrt{2\log(T\gamma_m^2) + \frac{8}{3}\gamma_m\log(T\gamma_m^2) + 6\gamma_m m\mathbb{E}[\tau]} \right)^2 \right\rceil \tag{11}
$$

**Lemma 9.** *Consider the delayed, aggregated, and anonymous feedback setting and the choice of $n_m$ given by equation 11. For any round $m \geq 1$ and any pair of active arms $(i,j) \in \mathcal{A}_m$, with probability at least $1 - \frac{3}{T\gamma_m^2}$,*

$$
\mu_{ij} - \bar{Y}_{ij}(m) \leq \frac{\gamma_m}{2}.
$$

*Proof.* We omit the proof here since the analysis of Lemma 1 in Pike-Burke et al. (2018) can be similarly applied here. $\square$

**Theorem 10.** *Under the delayed, aggregated, and anonymous feedback setting and the choice of $n_m$ given by equation 11, the expected regret $\mathbb{E}[\mathcal{R}(T))]$ of Algorithm 2 is upper bounded by*

$$
\sum_{i=2}^{K} O\left( \frac{K\log(T\Delta_i^2)}{\Delta_i} + K\log\left(\frac{1}{\Delta_i}\right)\mathbb{E}[\tau] \right)
$$

*Proof.* We follow the organization in the proof of Theorem 3 in Appendix B.2. Similarly, for any arm $i$, let $M_i$ be the round when arm $i$ is removed and define $m_i := \min\left\{ m \geq 1 : \gamma_m < \frac{2}{3}\Delta_i \right\}$. Then we know $\frac{\Delta_i}{3} \leq \gamma_{m_i} < \frac{2}{3}\Delta_i$. Additionally, we use the same notation of $N_{ij}$, $N_i$, $\mathcal{R}(T)$, and $\mathcal{R}_i(T)$ as employed in the proof of Theorem 3 in Appendix B.2. We once again break down the expected regret and bound each

element using Lemma 11 and 12.

$$\mathbb{E}[\mathcal{R}(T)] = \mathbb{E}\left[\sum_{i=1}^{K} \mathcal{R}_i(T)\right]$$

$$= \underbrace{\mathbb{E}\left[\sum_{i=1}^{K} \mathcal{R}_i(T)\mathbb{I}(M_1 \geq m_i)\right]}_{(a)} + \underbrace{\mathbb{E}\left[\sum_{i=1}^{K} \mathcal{R}_i(T)\mathbb{I}(M_1 < m_i)\right]}_{(b)}$$

**Lemma 11.** *For any arm $i \neq 1$,*

$$\mathbb{P}(M_i > m_i \text{ and } M_1 \geq m_i) \leq \frac{3}{T\gamma_{m_i}^2}.$$

*Proof.* The proof is identical to that of Lemma 7, with the only difference being the utilization of the upper bound probability from Lemma 9. □

**Lemma 12.** *Let the event $F_i(m)$ be the event that the optimal arm 1 is eliminated by arm $i$ in round $m$. Then, for any arm $i$,*

$$\mathbb{P}(F_i(m)) \leq \frac{3}{T\gamma_m^2}.$$

*Proof.* The proof is identical to that of Lemma 8, differing only in the upper bound probability from Lemma 9. □

**Bounding Term $(a)$.** By Lemma 11, we have

$$\mathbb{E}\left[\sum_{i=1}^{K} \mathcal{R}_i(T)\mathbb{I}(M_1 \geq m_i)\right]$$

$$\leq \sum_{i=1}^{K} \mathbb{E}\left[\mathcal{R}_i(T)\mathbb{I}(M_i \leq m_i)\right] + \sum_{i=1}^{K} \mathbb{E}\left[\frac{T\Delta_i}{2}\mathbb{I}(M_i > m_i, M_1 \geq m_i)\right]$$

$$\leq \sum_{i=1}^{K} \frac{\Delta_i}{2} n_{m_i} K + \frac{T}{2} \sum_{i=1}^{K} \frac{3\Delta_i}{T\gamma_{m_i}^2}$$

$$\leq \sum_{i=1}^{K} \frac{\Delta_i}{2} n_{m_i} K + \frac{3}{2} \sum_{i=1}^{K} \frac{9}{\Delta_i} = \sum_{i=1}^{K} \left(\frac{27}{2} \cdot \frac{1}{\Delta_i} + \frac{\Delta_i n_{m_i} K}{2}\right)$$

**Bounding Term** (b). Let $m_{\max} = \max_{i \neq 1} m_i$. Using Lemma 11 and 12, we obtain

$$\mathbb{E}\left[\sum_{i=1}^{K} \mathcal{R}_i(T)\mathbb{I}(M_1 < m_i)\right]$$

$$= \sum_{m=1}^{m_{\max}} \mathbb{E}\left[\mathbb{I}(M_1 = m) \sum_{i:m_i > m} \frac{1}{2} N_i \Delta_i\right]$$

$$\leq \sum_{m=1}^{m_{\max}} \mathbb{E}\left[\mathbb{I}(M_1 = m) T \max_{i:m_i > m} \Delta_i\right]$$

$$\leq \sum_{m=1}^{m_{\max}} 3\mathbb{P}(M_1 = m) T \gamma_m$$

$$\leq \sum_{m=1}^{m_{\max}} \left(\sum_{i:m_i < m} 3\mathbb{P}(M_1 \geq m_i, M_i > m_i) T \gamma_m + \sum_{i:m_i \geq m} 3\mathbb{P}(F_i(m)) T \gamma_m\right)$$

$$\leq \sum_{m=1}^{m_{\max}} \left(\sum_{i:m_i < m} \frac{9}{T\gamma_{m_i}^2} T \frac{\gamma_{m_i}}{2^{m-m_i}} + \sum_{i:m_i \geq m} \frac{9}{T\gamma_m^2} T \gamma_m\right)$$

$$\leq \sum_{i=1}^{K} \sum_{m=m_i}^{m_{\max}} \frac{9}{\gamma_{m_i}} 2^{m_i - m} + \sum_{i=1}^{K} \sum_{m=1}^{m_i} \frac{9}{2^{-m}}$$

$$\leq \sum_{i=1}^{K} \frac{54}{\Delta_i} + \sum_{i=1}^{K} 18 \cdot 2^{m_i} \leq \sum_{i=1}^{K} \frac{54}{\Delta_i} + \sum_{i=1}^{K} \frac{54}{\Delta_i} = \sum_{i=1}^{K} \frac{108}{\Delta_i}$$

where we also employed the relation $2^{m_i} = \frac{1}{\gamma_{m_i}} \leq \frac{3}{\Delta_i}$.

Consequently, by the upper bound proved for terms (a) and (b), the expected regret can be upper bounded by:

$$\mathbb{E}[\mathcal{R}(T)] \leq \sum_{i=1}^{K} \left(\frac{243}{2}\frac{1}{\Delta_i} + \frac{\Delta_i n_{m_i} K}{2}\right)$$

and since $n_{m_i}$ defined in equation 11 can be bounded as

$$n_{m_i} = \left\lceil \frac{1}{\gamma_{m_i}^2}\left(\sqrt{2\log(T\gamma_{m_i}^2)} + \sqrt{2\log(T\gamma_{m_i}^2) + \frac{8}{3}\gamma_{m_i}\log(T\gamma_{m_i}^2) + 6\gamma_{m_i}m_i\mathbb{E}[\tau]}\right)^2 \right\rceil$$

$$\leq \left\lceil \frac{1}{\gamma_{m_i}^2}\left(8\log(T\gamma_{m_i}^2) + \frac{16}{3}\gamma_{m_i}\log(T\gamma_{m_i}^2) + 12\gamma_{m_i}m_i\mathbb{E}[\tau]\right)\right\rceil$$

$$\leq 1 + \frac{8\log(4T\Delta_i^2/9)}{\gamma_{m_i}^2} + \frac{16\log(4T\Delta_i^2/9)}{3\gamma_{m_i}} + \frac{12\log_2(3/\Delta_i)\mathbb{E}[\tau]}{\gamma_{m_i}}$$

$$\leq 1 + \frac{72\log(4T\Delta_i^2/9)}{\Delta_i^2} + \frac{16\log(4T\Delta_i^2/9)}{\Delta_i} + \frac{72\log(3/\Delta_i)\mathbb{E}[\tau]}{\Delta_i},$$

we conclude that the total expected regret is bounded by

$$\mathbb{E}[\mathcal{R}(T)] \leq \sum_{i=2}^{K} \left(\frac{36K\log(4T\Delta_i^2/9)}{\Delta_i} + 8K\log(4T\Delta_i^2/9) + 36K\log(3/\Delta_i)\mathbb{E}[\tau] + \frac{243}{2}\frac{1}{\Delta_i} + \frac{1}{2}K\Delta_i\right).$$

$\square$

## D Lower Bound on the Regret

**Proposition 13.** *For any dueling bandit algorithm and $T$, there exists a $K$-armed dueling bandit such that $\mathcal{R}(T) \geq c\sqrt{TK/\tau_M}$, where $c > 0$ is a constant.*

*Proof.* The proof of this proposition closely follows the proof of Theorem 3 in (Vernade et al., 2020). Firstly, let's define a parameter $\mu^1 \in \mathbb{R}^{K \times K}$. For all $i > 1$, we set $\mu^1_{1i} = \frac{1}{2} + \Delta$ and $\mu^1_{i1} = \frac{1}{2} - \Delta$, where $0 \leq \Delta \leq \frac{1}{8}$. Additionally, for all $i \neq 1$ and $j \neq 1$, we define $\mu^1_{ij} = \frac{1}{2}$. Now, let $k = \arg\min_{i>1} \mathbb{E}_{\mu^1}[N_i(T)]$. Since we choose two arms at each time step, using the pigeonhole principle, we find that $\mathbb{E}_{\mu^1}[N_k(T)] \leq \frac{2T}{K-1}$. Next, we introduce another parameter $\mu^2 \in \mathbb{R}^{K \times K}$. We define $\mu^2_{k1} = \frac{1}{2} + \Delta$ and $\mu^2_{ki} = \frac{1}{2} + 2\Delta$ for all $i > 2$, while for all $i \neq k$ and $j \neq k$, $\mu^2_{ij} = \mu^1_{ij}$.

Then, by the definition of regret and $\mu^1$, it follows that

$$\mathcal{R}_{\mu^1}(T) \geq T\Delta - \frac{\Delta}{2}\mathbb{E}_{\mu^1}[N_1(T)]$$

$$= T\Delta - \frac{\Delta}{2}\mathbb{E}_{\mu^1}[N_1(T)\mathbb{I}(N_1(T) \leq T) + N_1(T)\mathbb{I}(N_1(T) > T)]$$

$$\geq T\Delta - \frac{T\Delta}{2}\mathbb{P}_{\mu^1}(N_1(T) \leq T) - T\Delta\mathbb{P}_{\mu^1}(N_1(T) > T)$$

$$= \frac{T\Delta}{2}\mathbb{P}_{\mu^1}(N_1(T) \leq T).$$

Similarly, we have

$$\mathcal{R}_{\mu^2}(T) \geq \frac{\Delta}{2}\mathbb{E}_{\mu^2}[N_1(T)] \geq \frac{T\Delta}{2}\mathbb{P}_{\mu^2}(N_1(T) > T).$$

Then, we use Bretagnolle-Huber inequality and obtain $\mathcal{R}_{\mu^1}(T) + \mathcal{R}_{\mu^2}(T) \geq \frac{T\Delta}{4}\exp(-KL(\mathbb{P}_{\mu^1}, \mathbb{P}_{\mu^2}))$. Also, the relative entropy between $\mathbb{P}_{\mu^1}$ and $\mathbb{P}_{\mu^2}$ can be expressed as follows:

$$KL(\mathbb{P}_{\mu^1}, \mathbb{P}_{\mu^2}) = \sum_{i=2}^{K} \mathbb{E}_{\mu^1}[N_{ki}(T)]d\left(\frac{\tau_m}{2}, \tau_m\left(\frac{1}{2} + 2\Delta\right)\right)$$

$$+ \mathbb{E}_{\mu^1}[N_{k1}(T)]d\left(\tau_m\left(\frac{1}{2} - \Delta\right), \tau_m\left(\frac{1}{2} + \Delta\right)\right)$$

$$+ \sum_{i=2}^{K} \mathbb{E}_{\mu^1}[N_{ik}(T)]d\left(\frac{\tau_m}{2}, \tau_m\left(\frac{1}{2} - 2\Delta\right)\right)$$

$$+ \mathbb{E}_{\mu^1}[N_{1k}(T)]d\left(\tau_m\left(\frac{1}{2} + \Delta\right), \tau_m\left(\frac{1}{2} - \Delta\right)\right)$$

$$\leq \mathbb{E}_{\mu^1}[N_1(T)] \cdot 32\tau_m\Delta^2 \leq \frac{64T}{K-1}\tau_m\Delta^2,$$

where $d(p,q) = p\log(\frac{p}{q}) + (1-p)\log(\frac{1-p}{1-q})$ and the inequality holds by the relation between the relative entropy and $\chi^2$ distance. Therefore,

$$\mathcal{R}_{\mu^1}(T) + \mathcal{R}_{\mu^2}(T) \geq \frac{T\Delta}{4}\exp\left(-\frac{64T}{K-1}\tau_m\Delta^2\right).$$

Finally, by setting $\Delta = \sqrt{\frac{K-1}{128T\tau_m}}$, we achieve

$$\mathcal{R}_{\mu^1}(T) + \mathcal{R}_{\mu^2}(T) \geq c\sqrt{TK/\tau_m},$$

which completes the proof. $\qquad\square$

