# OpenReview forum: "Biased Dueling Bandits with Stochastic Delayed Feedback"
_TMLR — Accepted by TMLR_

### Review · Reviewer_oe1e · 2024-07-04

**Summary Of Contributions:**

This paper studies dueling bandits under delayed feedback, where each delay is randomly drawn from a distribution that is independent of the chosen pair of arms and supported on $\mathbb{N}$. In particular, the paper assumes that when a comparison feedback between two arms is not yet available due to a delay, the learner presumes that one of the arms is preferred, resulting in a preference bias; one given example is that, in e-commerce, if a user is looking at an item A, and the platform recommends another item B, a delay suggests that the user stays at A and therefore presumably prefers A over B.

The paper proposes two algorithms: RUCB-Delay, for the case where the delay distribution is known and that any feedback can only be seen if the delay is no more than $M$ rounds, and MRR-DB-Delay which only requires that the expected delay is bounded and known. Regret guarantees for these algorithms are provided as well as an empirical study that uses preference datasets, to which simulated delays are added.

**Audience:**

Yes

**Broader Impact Concerns:**

No particular concerns. The authors could briefly discuss how this work can be potentially related to reinforcement learning with human feedback for large language models.

**Claims And Evidence:**

Yes

**Requested Changes:**

1. It would be helpful if the authors could discuss, for each choice of a pair of arms $(i,j)$ by the player, when/how the order of the arms matters because of comparative/preference bias.

2. In Section 3.1, it would be helpful if the authors could provide some explanations/intuitions about the definition of $S_ij$.

3. The latter half of Section 3.2 can be hard to follow -- some intuitive explanations would help?

4. In Section 4, there seems to be no discussion about the role of comparative/preference bias anymore.

5. In Section 5, it is mentioned that the baseline RUCB algorithm “uses the number of times arm $i$ beats arm $j$ to establish their upper confidence bound." Does this number include potentially delayed feedback?

**Strengths And Weaknesses:**

The problem studied in this paper, dueling bandits under delayed feedback, seems novel, and it leads to some unique challenges given that feedback is limited to relative comparisons. For the setting where the delay distribution is known, the proposed algorithm is adapted from RUCB and uses a modified upper confidence bound that manages delays; it recovers the optimal regret bound for dueling bandits when there is no delay. For the setting where only the expected delay is known, the proposed algorithm extends ideas from (Pike-Burke et al., 2018) to dueling bandits. The empirical study is nice to have, and seems to validate the theoretical findings.

I have some questions/concerns about the problem formulation. It seems to me that the preference/comparative bias is introduced by the algorithm/formulation and is not necessarily inherent in dueling bandits itself. The authors mention that in multi-armed bandits under delayed feedback (Vernade et al., 2017; 2020), the learner observes a zero reward when the reward is not yet available. I am not entirely convinced that this should be kept the same for dueling bandits, as this naive extension would imply that a learner should assume that one arm in a pair is primary, which does not seem native to the dueling bandits setting. It is not yet clear to me how a simple algorithm would behave if it simply does not use comparisons that have not been observed yet (e.g., Joulani et al., 2013, Section 4), instead of presuming that one arm is preferred by default. In particular, how would this baseline algorithm behave in the experiments?

In addition, the paper assumes that the delay distribution is independent of the selected pair of arms. In the context of preference comparisons, it would be a lot more interesting and natural to see settings where, for example, a user hesitates more because $\mu_{ij}$ is close to $1/2$.

The presentation of this paper seems fair. Some of the technical discussions (e.g., Section 3.2) can be a bit hard to follow.

---

> ### Author Response · Authors · 2024-07-27
>
> Thank you for your insightful comments and thorough review. We are pleased to hear that you find our work novel. Additionally, we appreciate your positive feedback on our theoretical findings and empirical study. Please see our detailed responses to your questions and concerns below.
>
>
> **Responses to Questions/Concerns:**
>
>
> 1. *It seems to me that the preference/comparative bias is introduced by the algorithm/formulation and is not necessarily inherent in dueling bandits itself. The authors mention that in multi-armed bandits under delayed feedback (Vernade et al., 2017; 2020), the learner observes a zero reward when the reward is not yet available. I am not entirely convinced that this should be kept the same for dueling bandits, as this naive extension would imply that a learner should assume that one arm in a pair is primary, which does not seem native to the dueling bandits setting.*
>
> We truly appreciate your suggestion. You are correct that the bias is not inherent to the dueling bandits themselves, but it arises if we assume a zero reward for unavailable rewards. Our intention was to convey that this setting is similar to those discussed in other bandit literature with binary rewards, such as the works by Vernade et al. (2017; 2020). As demonstrated in Figure 1, we find the challenge of preference or comparative bias more intriguing and relevant in practical scenarios. Addressing dueling bandits under delayed feedback without any bias would be a comparatively straightforward problem, which can also be effectively solved using our proposed algorithms. While the observation $Y_{s,t}$ may be biased, $\hat{\mu_{ij}}$, our estimator of $\mu_{ij}$ in Eq (4), is an unbiased estimator as shown in Proposition 1. We emphasize that our work is applicable to cases without any bias in dueling bandits.
>
>
> 2. *It is not yet clear to me how a simple algorithm would behave if it simply does not use comparisons that have not been observed yet (e.g., Joulani et al., 2013, Section 4), instead of presuming that one arm is preferred by default. In particular, how would this baseline algorithm behave in the experiments?*
>
> Thank you for your insightful comments. These days, we applied the method from Joulani et al. (2013) by only using the observed rewards in an additional experiment, and compared the regret performance of this method with our proposed algorithm, RUCB-Delay. The regrets are reported at time step 50,000, where the algorithms have converged. We assume the delay distribution follows a geometric distribution with p=0.01. We report the average cumulative regret across 10 runs. The results, shown in the table, demonstrate that RUCB-Delay effectively solves the dueling bandit problem with delayed feedback even in the absence of bias. Moreover, the results indicate that RUCB-Delay outperforms the method from Joulani et al. (2013).
>
> |                       | Six Rankers | MSLR | Tennis |
> |-----------------------|-------------|------|--------|
> | RUCB-Delay            | 1378.73        | 397.92  | 1262.63   |
> | Joulani et al. (2013) | 1603.38       | 471.04  | 1478.24   |
>
>
>
>
>
> 3. *In addition, the paper assumes that the delay distribution is independent of the selected pair of arms. In the context of preference comparisons, it would be a lot more interesting and natural to see settings where, for example, a user hesitates more because $\mu_{ij}$ is close to $1/2$.*
>
> Thank you for proposing a highly interesting and meaningful research direction. Upon further examination, we have found that our analysis can be extended to scenarios where the delay distribution depends on the pair of arms. For RUCB-Delay, we first define $\tau_t^{ij}$ as the cumulative distribution function of the delay distribution for a pair of arms $(i,j)$. Then, we modify $\tau_t$ to $\tau_t^{ij}$ in the definitions of $S_{ij}(t)$ and $\tilde{N_{ij}}(t)$. Proposition 1, Lemma 2, and Lemma 3 remain valid since they pertain to each pair of arms individually. Consequently, the regret analysis (Theorem 1) also holds with minor modifications. The upper bound in Theorem 1 should be adjusted as shown below. Theorem 2 should be similarly updated.
> $$O \left(\frac{MK^2}{\delta} \right) + \tilde{O} \left(  \sum_{i <j} \frac{\alpha}{ (\tau_1^{ij})^2 \min ( \Delta_i^2, \Delta_j^2)}\right)  + O \left( \sum_{j=2}^K \frac{\alpha}{(\tau_1^{1j})^2 \Delta_j^2 } \log T \right)$$
> For MRR-DB-Delay, we define $\mathbb{E}[D^{ij}]$ as the expected value of the delay when selecting arms $(i,j)$. By replacing $\mathbb{E}[D]$ with $\mathbb{E}[D^{ij}]$ in Equation (6), Lemma 4 remains valid. Additionally, Theorem 3's bound should be updated by substituting $\mathbb{E}[D]$ with $\mathbb{E}[D^{ij}]$.
>
> While our analysis can be extended to scenarios where the delay distribution is not independent, we assume independence in our work for simplicity and clarity. We have included a footnote on page 4 (Section 2) addressing this extension.

---

> > ### Author Response · Authors · 2024-07-27
> >
> > **Responses to Requested Changes:**
> >
> >
> > We have provided detailed, point-by-point responses to the reviewer, with pointers to the specific sections in the manuscript where changes were made.
> >
> > 1. *It would be helpful if the authors could discuss, for each choice of a pair of arms $(i,j)$ by the player, when/how the order of the arms matters because of comparative/preference bias.*
> >
> > Thank you for your insightful comments. If the reward for an action is delayed and therefore unobserved, it is treated as a zero reward in our framework. The bias naturally happens when we assume zero reward for unobserved reward. This assumption impacts the order of arms, particularly when the reward is delayed. Moreover, this treatment naturally gives an advantage to the second arm in a pair. For example, if arm $i$ is presented first and arm $j$ second, and the reward is delayed, the player observes a zero value while making a decision. This results in arm $j$ being favored, while arm $i$ suffers from the delay-induced bias. We put a detailed discussion of this topic into our revision in Section 1.
> >
> >
> > 2. *In Section 3.1, it would be helpful if the authors could provide some explanations/intuitions about the definition of $S_{ij}$.*
> >
> > Thank you for the suggestion. $S_{ij}(t)$ captures the bias-corrected count of arm $i$ beating arm $j$ up to time $t$. When comparing $(i,j)$ and if the current observation implies $j > i$, we add $\tau$ instead of 1. This adjustment accounts for the preference bias favoring the second arm over the first arm, which enables us to construct an unbiased estimator of $\mu_{ij}$. We have added explanations regarding the definition of $S_{ij}$ in Section 3.1.
> >
> >
> >
> > 3. *The latter half of Section 3.2 can be hard to follow -- some intuitive explanations would help?*
> >
> > Thank you for the comments. At each time step, we define a potential champion set $\mathcal{C}$ consisting of arms that optimistically win against all other arms, meaning $U_{i j} \geq \frac12$ for all $j$. We then update the current best arm set $\mathcal{B}$, which will either have one element or be empty. The idea is twofold: first, the arm in $\mathcal{B}$ loses its top position as the best arm if it is optimistically beaten by another arm, and second, the arm $u_t$ will be selected from $\mathcal{B}$ with high probability, or from the potential champion set $\mathcal{C}$. We have revised and enhanced the explanation in Section 3.2 for greater clarity.
> >
> >
> > 4. *In Section 4, there seems to be no discussion about the role of comparative/preference bias anymore.*
> >
> > In Section 4, we presented an algorithm designed for situations where the expected delay is known. We focused on explaining the algorithm itself and providing its regret analysis. The proposed method automatically manages the bias issue. We note that establishing a regret upper bound involved successfully addressing the preference bias. By decomposing the sum of discrepancies between $\mu_{ij}$ and the observation $Y_{s,t_m}$, we were able to separately analyze the effect of preference bias. We have added this explanation to Section 4.2.
> >
> > 5. *In Section 5, it is mentioned that the baseline RUCB algorithm “uses the number of times arm $i$ beats arm $j$ to establish their upper confidence bound." Does this number include potentially delayed feedback?*
> >
> > Yes, it does include potentially delayed feedback. For biased dueling bandit problems, we assume zero rewards for unobserved outcomes. Therefore, we incorporated potentially delayed feedback when calculating the number of rewards received. The experimental results in Section 5 show that our novel estimator ensures the algorithm successfully converges and handles both delay and bias issues. Even when considering an unbiased scenario, our results, as shown in the table in response to item 2, demonstrate that our algorithm outperforms the baseline RUCB algorithm when it does not account for potentially delayed feedback. We have added an explanation about this comment in Section 5.
> >
> > Thank you for your valuable comments and suggestions, which definitely help us improve the quality of our work.

---

### Review · Reviewer_WpS9 · 2024-07-17

**Summary Of Contributions:**

This paper investigates the dueling bandit problem with stochastic delayed feedback. The authors introduce two algorithms designed to address scenarios where either the full delay distribution information is available or only the expected delay is known. They provide a comprehensive regret analysis for the proposed algorithms and demonstrate their empirical performance using both synthetic and real datasets.

**Audience:**

Yes

**Broader Impact Concerns:**

this is a theory paper and the reviewers do not see any concerns.

**Claims And Evidence:**

Yes

**Requested Changes:**

See the weaknesses section above.

**Strengths And Weaknesses:**

**Strengths**

* This paper addresses a novel and intuitive dueling bandit problem with stochastic delayed feedback.

* For the setting with known delay distribution, the authors introduce a modified upper confidence bound (UCB) algorithm. This modification could potentially benefit other UCB-based dueling bandit algorithms dealing with delayed feedback.

* For the setting with known expected delay, they propose a multi round-robin algorithm and provide a thorough regret analysis.

* Experiments on six datasets demonstrate the superior performance of the proposed algorithms.

**Weaknesses**

* Despite the example provided in Fig. 1, it is still unclear why unobserved means A>B in dueling bandits: if a user has not yet decided between two recommended items, why do we assume the user prefers the first one?

* More explanation is needed to justify the practicality of this model.

* The authors demonstrate that RUCB-Delay achieves the optimal regret bound for the dueling bandit problem when there is no delay. However, they do not provide regret lower bounds for settings with known delay distribution or known expected delay, leaving the optimality of the proposed algorithms in these settings uncertain.

* This paper studies the known expected delay setting. In contrast, prior literature on stochastic bandits with delayed feedback has explored settings without prior delay information. This limitation may narrow the scope of the paper.

* Experiments: The authors do not explain why RR-DB-Delay outperforms RUCB-Delay in Figure 2(c). It would be helpful to include experiments on different delay distributions to better understand the algorithms' performance across varied conditions.

---

> ### Author Response · Authors · 2024-07-27
>
> Thank you very much for your valuable feedback. We are delighted that you found our problem setting both novel and intuitive. We are also pleased to know that you appreciated the thorough regret analysis of our algorithm and its potential extension to other UCB-based bandit algorithms. Below, we have provided detailed responses to each of your comments.
>
> 1 & 2. We find the biased dueling bandit problem to be highly practical with various real-world applications. For example, in Figure 1, we consider a scenario where a user is browsing product A, and the platform suggests product B in the sidebar. In real-world situations, there will be delays due to the user contemplating the decision or random network delays. We can only observe that the user remains on product A over B, which means we can only infer that B cannot beat A until the end of the delay. This makes the biased dueling bandit problem both practical and important to study. The system cannot distinguish whether a user is still contemplating which item to choose or has decided not to choose either item. To ensure efficient learning, we replace the delayed feedback with a suitable value. It is crucial to fully leverage user actions on the platform to maximize the effectiveness of the learning process. Even if the user did not choose either product, there might still be a subtle preference.
>
>
> While obtaining relative preference feedback is a reliable method for consistently gathering accurate user preferences, it is not always favored by users on e-commerce or video platforms. For instance, in a video recommendation system like Netflix, a traditional dueling bandit approach would require the system to present two videos side-by-side and ask the user which one they prefer. This method is time-consuming and demands extra actions from users. Additionally, repeatedly requesting feedback can decrease user satisfaction with the platform. Because real world applications will not wait for users, user always take time to make decision or pairwise comparison. Delay is inevitable in applications.
>
> In contrast, situations like in Figure 1(a) can be naturally observed through user actions. Unlike the traditional method, tracking user actions naturally gathers abundant data, making it a more practical approach. The traditional dueling bandit approach, on the other hand, requires additional time and effort from users, making it less efficient for data collection. This is one of the central motivations behind this paper. By analyzing user actions, we aim to extract valuable preference information to refine the recommendation system, ultimately enhancing user experience and engagement.

---

> ### Author Response · Authors · 2024-07-27
>
> 3. We fully acknowledge the importance of proving a lower bound for settings with delayed feedback. However, deriving such lower bounds is highly challenging within the bandit literature. Despite this difficulty, we did provide a non-asymptotic lower bound in the delayed feedback setting, as mentioned in Remark 2. We included it as a remark because we are uncertain about its tightness. Additionally, we believe our experimental results clearly demonstrate the superior performance of our proposed algorithms.
>
>
>
>
> 4. We acknowledge the limitation of assuming prior delay information. However, to the best of our knowledge, our work is the first to address dueling bandit problems in a delayed feedback setting, providing a solid foundation for future research. As a next step, we aim to develop an algorithm that does not rely on this assumption, specifically for biased dueling bandit settings. Additionally, we plan to explore methods for estimating delay information and analyzing its impact on regret. This will further enhance the applicability and robustness of our methods. Nevertheless, we would like to point out that many papers on delayed feedback make similar assumptions about the delay distribution. To mention a few, we list some papers that study delayed feedback along with their assumptions regarding the delay distribution.
> Stochastic Bandit Models for Delayed Conversions (UAI 2017): Known full delay distribution
> Bandits with Delayed, Aggregated Anonymous Feedback (ICML 2018): Known and bounded expected delay
> Delayed Feedback in Kernel Bandits (ICML 2023): Known expected delay and sub-exponential
> Delayed Feedback in Generalised Linear Bandits Revisited (AISTATS 2023): Sub-exponential delay
>
>
> 5. The intuitive reason why RR-DB-Delay generally has higher regret performance than RUCB-Delay is that it initially tests all possible arm pairs. However, if the problem is relatively easy to solve (i.e., $\Delta_{i}$ is large for $i \neq 1$), the suboptimal arms will be eliminated in the early stages, allowing RR-DB-Delay to converge faster than RUCB-Delay. We have observed that, although RR-DB-Delay initially shows worse performance, it converges faster than RUCB-Delay and ultimately achieves better results.
>
>     We conducted additional experiments using a different delay distribution. While our experiments in Section 5 assumed delays followed a geometric distribution, we also tested with delays following a normal distribution $N(100, 25)$ here. We kept the mean value of delay consistent with Section 5 to enable a fair comparison. We observe that the results were quite similar across both distributions.
>
> |              | Arithmetic | Car Preference | Sushi   |
> |--------------|------------|----------------|---------|
> | RUCB-Delay   | 2109.79    | 1656.99        | 3774.82 |
> | RR-DB-Delay  | 3826.94    | 2465.12        | 5784.33 |
> | MRR-DB-Delay | 5011.32    | 2896.55        | 7320.64 |
>
>
>
>
> We sincerely appreciate your thorough review and insightful feedback. We are eager to engage in further discussions with you.

---

### Review · Reviewer_5bEz · 2024-07-18

**Summary Of Contributions:**

This paper studies the stochastic dueling bandits problem with stochastic delay. Here, the number of arms is finite, and the distribution or expected value of stochastic delays is assumed to be known. The goal is to learn a policy that selects two arms (duel) in each round to minimize the cumulative regret (dueling bandits variant).

The authors propose two algorithms: RUCB-Delay when the distribution of delays is known and MRR-DB-Delay when the only expected value of delays is known. They have shown sub-linear regret bounds for both algorithms and corroborated their theoretical results using experiments on instances derived from real-world datasets.

**Audience:**

Yes

**Broader Impact Concerns:**

Since it is a theoretical paper, I do not find any ethical concerns.

**Claims And Evidence:**

Yes

**Requested Changes:**

Please address the weaknesses mentioned in **Strengths And Weaknesses**.

**Strengths And Weaknesses:**

#### **Strengths of the paper:**
1. To the best of my knowledge, this paper is the first to consider the problem of stochastic dueling bandits with stochastic delays.
2. The authors consider two settings: one where the distribution of delays is known and a second where only the expected value of delays is known. The authors propose algorithms for both settings and show that both algorithms have sub-linear regret3
3. Finally, the authors have considered different problem instances derived from real datasets to show the performance gain of the proposed algorithms over baselines.


#### **Weaknesses of the paper:**
1. Assumption: The assumption of censored feedback is a bit strict in unobserved feedback. Saying the unobserved feedback only implies A>B will cause the biased estimation of preference feedback because it is possible that a user may not like both A and B in real life.

2. Motivational examples and delay assumption: There needs to be a motivational example that fits well into your setting. The given example is more suitable for contextual dueling bandits. It is unclear how one can know the distribution or expected value of delay in the motivational example considered in the paper. The assumptions considered on delays limit the applications of proposed algorithms.

3. Analysis novelty: The dueling bandits and bandits problem with stochastic delay are now separately well-studied problems. From the paper, it is unclear what the key challenges are with bringing stochastic delay feedback in stochastic dueling bandits. The settings considered in the paper are very simple compared to the stochastic delay feedback settings already considered in bandits. The best problem setting can be the most settings (e.g., contextual dueling bandits with non-linear reward function and unknown stochastic delays), which can be motivated from the following paper:

    1. Bayesian Optimization under Stochastic Delayed Feedback: https://proceedings.mlr.press/v162/verma22a/verma22a.pdf

    2. Delayed feedback in kernel bandits: https://proceedings.mlr.press/v202/vakili23a/vakili23a.pdf

---

> ### Author Response · Authors · 2024-07-27
>
> Thank you very much for your insightful comments and careful review. We appreciate the opportunity to clarify our assumptions and the applicability of our algorithm. Please see our responses to your concerns:
>
>
>
> 1. In Figure 1, we consider an example where a user is browsing product A, and the platform suggests product B in the sidebar. In real-world situations, there will be delays due to the user contemplating the decision or random network delays. We can only observe that the user remains on product A over B, which means we can infer that B can not beat A. It is crucial to fully leverage user actions on the platform to maximize the effectiveness of the learning process. To ensure efficient learning, we replace the delayed feedback with a suitable value. As the reviewer mentioned, in real life, a user may not prefer either A or B enough to select one. However, they might still have a preference between the two and in the context of dueling bandit problems, we assume an underlying stochastic pairwise preference model. The goal is to accurately estimate this model. While the observation $Y_{s,t}$ may be biased, $\hat{\mu_{ij}}$, our estimator of $\mu_{ij}$ in Eq (4) of $\mu_{ij}$ in Eq (4) is an unbiased estimator as shown in Proposition 1. This new estimator leads to successful performance in Section 5. We emphasize that our work is applicable to cases where there is no bias in dueling bandits.
>
>
> 2. Our example aims to introduce the new concept of comparative/preference bias in dueling bandits and to explain its origins and why it occurs. Our algorithms are specifically designed to address both this bias and the problem of delayed feedback at the same time. While our current focus is on these challenges, we acknowledge that exploring contextual dueling bandits presents an interesting and valuable direction for future research. This could further enhance our understanding and handling of user preferences in some dynamic real-world environments.
>
>     We also acknowledge the limitation of assuming prior delay information. To the best of our knowledge, our work is the first to address dueling bandit problems in a delayed feedback setting, providing a solid foundation for future research. As a future research direction, we aim to develop an algorithm that rely on milder assumptions in delayed dueling bandit settings. Additionally, we plan to explore methods for estimating delay information and analyzing its impact on regret. Nevertheless, we would like to point out that many papers on delayed feedback, including the work (b.) mentioned by the reviewer, make similar assumptions about the delay distribution. To mention a few, we list some papers that study delayed feedback along with their assumptions regarding the delay distribution.
>
>     a. Stochastic Bandit Models for Delayed Conversions (UAI 2017): Known full delay distribution
>
>       b. Bandits with Delayed, Aggregated Anonymous Feedback (ICML 2018): Known and bounded expected delay
>
>       c. Delayed Feedback in Kernel Bandits (ICML 2023): Known expected delay and sub-exponential
>
>       d. Delayed Feedback in Generalised Linear Bandits Revisited (AISTATS 2023): Sub-exponential delay
>
>
>
>
> 3. Our study introduces and formulates a new biased dueling bandit framework that incorporates preference bias between selections. This setting goes beyond a simple combination of the traditional dueling bandit and delayed feedback settings. Our work is highly innovative as it addresses both the common challenges of delayed feedback and the practical concerns of bias simultaneously. In our paper, we developed two algorithms with comprehensive theoretical analysis, which includes a high-probability concentration inequality for the parameter $\mu_{ij}$ and a high-probability upper bound on regret.
>
>    Specifically, we introduced a series of new variables and proposed a novel upper confidence bound. This enabled us to develop an unbiased estimator for $\mu_{ij}$. Section 5 demonstrates that our novel estimator significantly enhances the algorithm’s effectiveness and robustness to delayed observations, as the baseline RUCB fails to converge across all datasets. We anticipate that many UCB-based bandit algorithms can transition to delayed feedback scenarios using our estimator and upper bound. We have illustrated how the RR-DB algorithm (Saha & Gaillard, 2022) can be modified in a similar manner in Remark 1.
>
>
>
> We deeply appreciate your thorough review and valuable insights. We look forward to engaging in further discussions with you.

---

### Decision · Action_Editor_CtmT · 2024-08-23

**Recommendation:** Accept with minor revision

**Comment:**

I would ask the authors to please revise the last claim in light of my comment in the Claims and Evidence section. I believe just a factual summary description of the experiments run is sufficient (even just removing the word "comprehensive" is fine with me).

**Audience:**

It is my and the reviewer's opinion that the paper will have an audience among the TMLR community.

**Claims And Evidence:**

The paper's main claims are justified by both theoretical guarantees in the form of proven theorems as well as numerical experiments.
The one claim I find to not be fully supported is the claim stating the paper contains a "comprehensive empirical evaluation" of the proposed algorithms. I believe this is a slight overstatement given that all proposed datasets are very small relative to what practitioners might encounter.